# Semaglutide ameliorates cardiac remodeling in male mice by optimizing energy substrate utilization through the Creb5/NR4a1 axis

Yu-Lan Ma[1,2,3], Chun-Yan Kong[1,2,3], Zhen Guo[1,2,3], Ming-Yu Wang[1,2], Pan Wang[1,2], Fang-Yuan Liu[1,2], Dan Yang[1,2], Zheng Yang[1,2] & Qi-Zhu Tang [1,2] ✉

Semaglutide, a glucagon-like peptide-1 receptor agonist, is clinically used as a glucose-lowering and weight loss medication due to its effects on energy metabolism. In heart failure, energy production is impaired due to altered mitochondrial function and increased glycolysis. However, the impact of semaglutide on cardiomyocyte metabolism under pressure overload remains unclear. Here we demonstrate that semaglutide improves cardiac function and reduces hypertrophy and fibrosis in a mouse model of pressure overload-induced heart failure. Semaglutide preserves mitochondrial structure and function under chronic stress. Metabolomics reveals that semaglutide reduces mitochondrial damage, lipid accumulation, and ATP deficiency by promoting pyruvate entry into the tricarboxylic acid cycle and increasing fatty acid oxidation. Transcriptional analysis shows that semaglutide regulates myocardial energy metabolism through the Creb5/NR4a1 axis in the PI3K/AKT pathway, reducing NR4a1 expression and its translocation to mitochondria. NR4a1 knockdown ameliorates mitochondrial dysfunction and abnormal glucose and lipid metabolism in the heart. These findings suggest that semaglutide may be a therapeutic agent for improving cardiac remodeling by modulating energy metabolism.

Chronic stress, encompassing conditions such as hypertension, is widely recognized for leading to adverse cardiac remodeling and subsequent heart failure (HF), a condition with high rates of mortality and morbidity[1]. Energy metabolic flexibility enables a healthy heart to adapt its utilization of metabolic substrates in response to fluctuations in circulating substrate concentrations and oxygen availability, ensuring sustained ATP production[2]. Unfortunately, the heart loses metabolic flexibility and undergoes myocardial metabolic remodeling at the onset of HF. While cardiac energy is generated primarily by mitochondrial oxidative phosphorylation, glycolysis contributes approximately 5% to the total energy production[3]. In healthy individuals, the heart derives approximately 60-90% of its energy from fatty acid oxidation[2]. In HF, the inadequate supply of oxygen causes the heart to take up and utilize glucose as a more efficient

energy substrate[4]. However, defects in mitochondrial structure and function lead to increased glucose uptake but incomplete oxidation, thereby inducing glycolysis[5]. Additionally, the transport and metabolism of pyruvate into the tricarboxylic acid (TCA) cycle are reduced in cases of HF[6,7]. Studies have shown that cardiac energy metabolism in HF patients switches from fatty acid oxidation to glycolysis over glucose oxidation, resulting in diminished ATP generating capacity and inadequate cardiac energy supply[8,9]. Thus, addressing the inadequate myocardial energy supply, primarily through mitigating mitochondrial deficits, is crucial for HF management.

Preclinical studies have demonstrated that compounds promoting glucose oxidation, such as dichloroacetate (DCA), can inhibit pyruvate dehydrogenase complex (PDC) kinase. This inhibition stimulates PDC activity, enhancing glucose oxidation and mitochondrial

[1]Department of Cardiology, Renmin Hospital of Wuhan University, Wuhan 430060, PR China. [2]Hubei Key Laboratory of Metabolic and Chronic Diseases, Wuhan 430060, PR China. [3]These authors contributed equally: Yu-Lan Ma, Chun-Yan Kong, Zhen Guo. ✉e-mail: qztang@whu.edu.cn

respiration, and subsequently mitigates cardiac hypertrophy in HF models using animals[10]. Unfortunately, the toxic effects of DCA have limited its clinical use and further development[11]. Another compound, elamipretide, appears to influence metabolism by augmenting mitochondrial function; however, preclinical evidence indicates it fails to enhance cardiac function in HF scenarios[12]. Consequently, identifying drugs that boost mitochondrial function and glucose oxidation to address the energy deficiency in HF represents a significant challenge[13]. Semaglutide (Sema), in contrast, has exhibited cardiovascular advantages in clinical trials, notably among HF patients with obesity and preserved ejection fraction[14]. Nevertheless, research into Sema's mechanistic effects on HF remains scant. Our study seeks to bridge this knowledge gap by exploring the detailed metabolic mechanisms underlying Sema's cardiovascular benefits.

Glucagon-like peptide-1 receptor agonists (GLP-1RAs) are widely used clinically as blood glucose-lowering agents in the treatment of type 2 diabetes mellitus (T2DM), and these drugs also have a weight-lowering effect[15]. Recently approved by the US Food and Drug Administration (FDA) as the first oral GLP-1RA for treating T2DM, Sema has demonstrated cardiovascular benefits in clinical trials involving patients with T2DM[16]. With its pronounced effect on weight loss, Sema has predominantly been adopted in clinical practice as a therapeutic agent for weight management[17]. In addition, Sema has been confirmed to have an ameliorative effect on disturbed lipid and glucose metabolism in obesity syndrome and T2DM[18,19]. In particular, Sema has recently been shown to have a significant improvement in HF with preserved ejection fraction (HFpEF) and obesity[14]. Despite these promising outcomes, the specific mechanisms driving Sema's cardiac benefits remain to be elucidated. Consequently, our study aims to explore Sema's impact on cardiac remodeling, aiming to underscore its therapeutic potential in addressing pathological cardiac remodeling and HF.

Our study revealed that Sema treatment effectively suppressed pressure overload-induced pathological hypertrophy and cardiac dysfunction in male mice, a condition accompanied by significant mitochondrial damage in chronic HF induced by transverse aortic constriction (TAC). This damage was characterized by a reduction in mitochondrial count, alongside mitochondrial swelling, fragmentation, and deterioration of the inner mitochondrial membrane and cristae. The metabolic profile of these mice shifted towards an increase in glycolysis and a significant decrease in fatty acid oxidative phosphorylation. Treatment with Sema reduced mitochondrial functional and structural abnormalities. Correspondingly, Sema promoted the entry of pyruvate into mitochondria, facilitated the complete oxidation of glucose, and increased the oxidation capacity of fatty acids to some extent. NR4a1 is a member of the nuclear receptor superfamily, which plays an important role in the control of metabolism, autophagy, inflammation, and mitochondrial biogenesis[20–22]. Transcriptome analysis disclosed that NR4a1 mitochondrial translocation is a major target in the regulation of glucose and lipid metabolism by Sema. NR4a1 plays a role in the regulation of mitochondria-dependent apoptosis[23]. Our findings suggest that NR4a1 expression and translocation after TAC are crucial factors in mitochondrial dysfunction, resulting in metabolic remodeling and structural remodeling of the heart. These findings highlight the significant role of Sema in regulating glycolipid metabolism to mitigate cardiac remodeling. In particular, we revealed the regulatory mechanism of the Sema critical regulator NR4a1.

## Results

### Sema ameliorates myocardial hypertrophy and cardiac dysfunction induced by TAC for eight weeks in mice
To investigate the effect of Sema on pathological cardiac remodeling, after TAC or sham surgery, we administered different doses (4 μg/kg, 12 μg/kg, and 60 μg/kg) of Sema or vehicle to mice every day for eight weeks (Fig. S1A). We found no significant effect of different doses of Sema on blood glucose, cholesterol or liver functions in mice (Fig. S1D–G). Sema is mainly used as a weight loss drug in clinical practice[24]. Therefore, we chose the high dose (60 μg/kg) with the most pronounced effect on body weight as the subsequent experimental dose (Fig. S1B–C). Our results showed that mice with TAC-induced cardiac dysfunction at eight weeks were significantly improved after Sema treatment, as evidenced by an elevated EF, FS, and a decreased LVIDd and LVIDs (Fig. 1A–D). In addition, hemodynamics significantly improved after Sema treatment, as shown by ±dp/dt, P-V loop indicators (Fig. 1E, F). TAC-induced chronic stress resulted in substantial increases in myocardial hypertrophy (ANP and BNP mRNA levels) and cardiomyocyte cross-sectional area (CSA); in addition, the heart weight to body weight ratio (HW/BW), the lung weight to body weight ratio (LW/BW) and heart weight to tibial length ratio (HW/TL) levels were significantly increased. In contrast, TAC-induced myocardial hypertrophy was significantly attenuated after the administration of Sema (Fig. 1G–J). These results suggest that Sema can ameliorate the pathological ventricular hypertrophy and dysfunction induced by TAC.

### Sema reverses pathological myocardial remodeling induced by TAC
Clinically, many patients come for treatment only when they already have altered cardiac function. Thus, we next further verified whether Sema reversed pathological myocardial remodeling by administering Sema treatment after four weeks after TAC (Fig. 2A). We found that cardiac dysfunction was reversed in mice treated with Sema compared to that before the administration of the drug, as shown by the B- and M-mode echocardiograms, and elevated EF and FS (Fig. 2B–D). Moreover, WGA and HE staining revealed that the CSA of cardiomyocytes from Sema treatment mice was smaller than mice in control group (Fig. 2E–G). Likewise, Sema treatment mitigated collagen deposition (PSR staining), as assessed by reduced collagen volume in hearts (Fig. 2H, I). In addition, the heart weight to body weight (HW/BW) and heart weight to tibial length (HW/TL) ratio were significantly reduced after Sema administration (Fig. 2J). Meanwhile, the mRNA levels of ANP, BNP, COL I, and COL III also indicated that Sema significantly attenuated TAC-induced myocardial hypertrophy and fibrosis (Fig. 2K). Parallel transcriptomic analysis of myocardial tissue from mice in the reversal experiment revealed that the treatment of Sema significantly rescued TAC-induced changes at the gene level (Fig. S2A). The reversal experiment showed significant enrichment of DEGs in extracellular matrix (ECM), cell migration, and blood vessel development pathways (Fig. S2B). It has been demonstrated that ECM plays an important role in pressure overload-induced heart failure, and excessive deposition of ECM proteins leads to fibrosis and deterioration of cardiac function[25]. Similarly, cell migration is crucial for fibrosis, with myofibroblasts under pressure overload conditions in HF showing enhanced migration abilities, secreting ECM proteins, and leading to myocardial remodeling and stiffening, thereby impairing cardiac function[26]. In addition, increased endothelial cell proliferation and promotion of angiogenesis significantly improve cardiac function in pressure overload-induced myocardial remodeling[27]. Sema exerts anti-fibrotic and anti-hypertrophic effects by attenuating ECM proteins (Fig. S2D), and, consistently, Sema application inhibits deterioration of cardiac function, development of cardiac fibrosis and cardiac hypertrophy in response to pressure overload (Fig. 2). These results suggest that Sema can also reverse TAC-induced pathological myocardial remodeling.

### Sema reduces TAC-induced mitochondrial fusion and division and maintains normal mitochondrial morphology and function
Mitochondria are the energy centre of the heart, and dynamic mitochondrial division and fusion regulate fuel oxidative phosphorylation

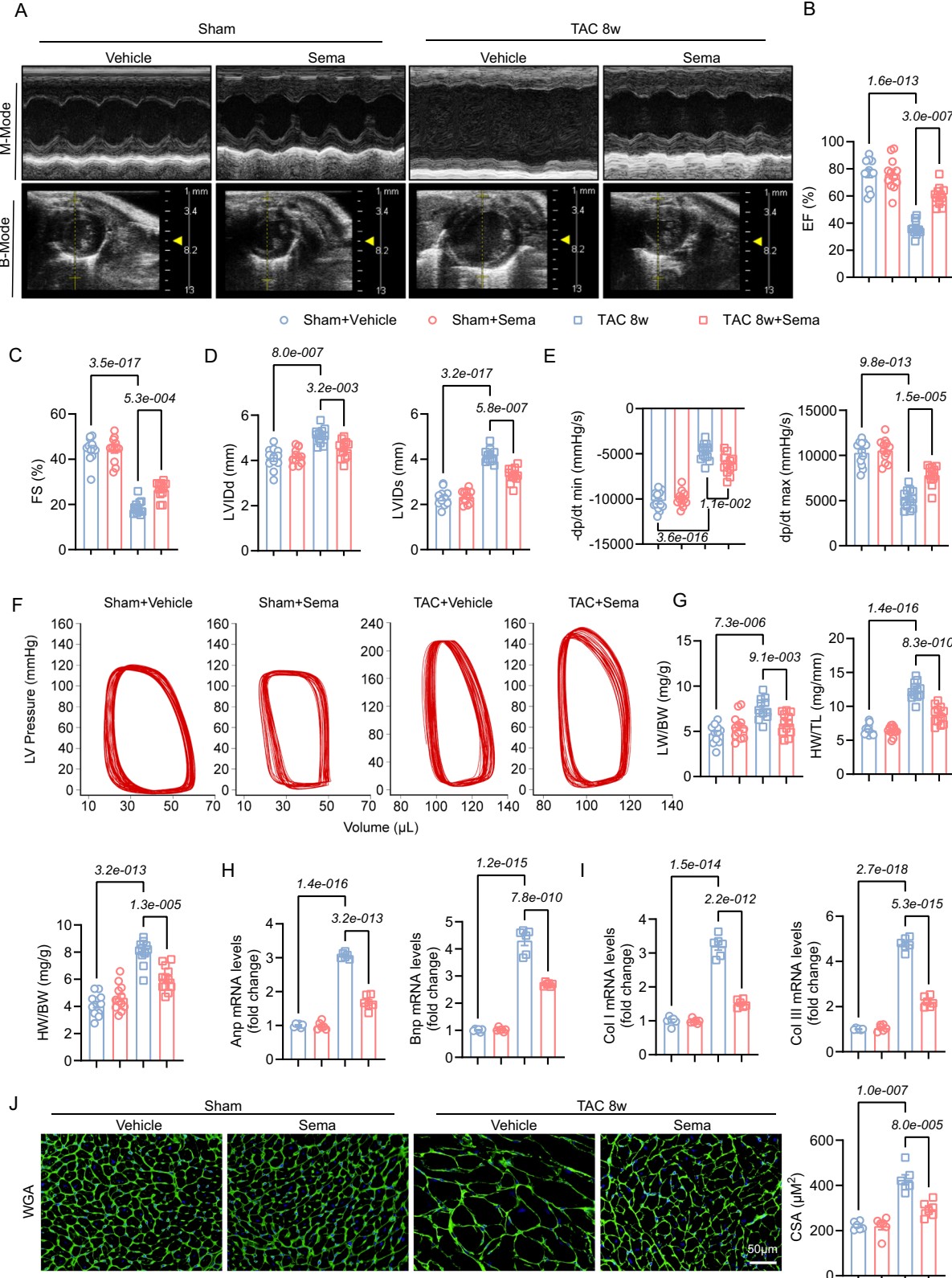

and the intracellular ATP supply in cardiomyocytes[28]. Transcriptome GSEA showed that 234/1366 upregulated gene sets were significant at FDR < 25% and enriched at nominal p value < 1% in the TAC 8w with Sema group; of these gene sets, 92 were related to the metabolism of carbohydrates, energy, lipid nucleotides, amino acids, and mitochondrial function (Fig. S3A, B; Supplementary Data 3). Thus, we hypothesized that Sema regulates the morphology and function of

mitochondria in pathological myocardial remodeling. Mitochondrial respiratory chain proteins comprise five enzymatic complexes and guarantee the production of ATP[29]. Transcriptome profiling showed the significant downregulation of most mitochondrial respiratory chain-related genes in hearts induced by TAC for eight weeks, whereas the hearts from the mice treated with Sema showed a reversed expression pattern (Fig. 3A). Next, we performed transmission

**Fig. 1 | Sema attenuates cardiac hypertrophy, fibrosis and cardiac dysfunction in mice that underwent TAC for eight weeks. A** Representative B- and M-mode echocardiographic imaging of the heart. **B**−**D** EF, FS, LVIDd and LVIDs were assessed by echocardiography ($n = 12$); EF: $F_{(3, 44)} = 54.15$, $P = 8.2e\text{-}015$; FS: $F_{(3, 44)} = 101.2$, $P = 9.1e\text{-}020$; LVIDd: $F_{(3, 44)} = 15.91$, $P = 3.8e\text{-}007$; LVIDs: $F_{(3, 44)} = 90.69$, $P = 7.2e\text{-}019$. **E** The maximum rate of isovolumetric systolic LV pressure increase ( + dp/dt max) and the minimum rate of isovolumetric diastolic LV pressure decrease (-dp/dt min) ($n = 12$); +dp/dt max: $F_{(3, 44)} = 52.79$, $P = 1.3e\text{-}014$; -dp/dt min: $F_{(3, 44)} = 88.28$, $P = 1.2e\text{-}018$. **F** The ventricular pressure-volume loop (P-V loop) ($n = 6$). **G** The ratio of LW, BW, HW, and TL ($n = 12$); HW/TL: $F_{(3, 44)} = 83.11$, $P = 3.7e\text{-}018$; HW/BW: $F_{(3, 44)} = 47.17$, $P = 8.5e\text{-}014$; LW/BW: $F_{(3, 44)} = 11.35$, $P = 1.2e\text{-}005$. **H**, **I** The mRNA expression of ANP, BNP, collagen I and collagen III in the heart

($n = 6$); ANP: $F_{(3, 20)} = 336.5$, $P = 2.8e\text{-}017$; BNP: $F_{(3, 20)} = 276.1$, $P = 1.9e\text{-}016$; COL 1: $F_{(3, 20)} = 210.4$, $P = 2.7e\text{-}015$; COL3: $F_{(3, 20)} = 495.5$, $P = 6.3e\text{-}019$. **J** Myocardial hypertrophy was detected by WGA staining and then quantitatively analyzed (scale bar: 50 μm; $n = 6$); CSA: $F_{(3, 20)} = 36.38$, $P = 2.7e\text{-}008$. All results are shown as the mean ± SEM and analyzed by one-way ANOVA followed by Bonferroni post hoc test (**B**−**E** and **G**−**J**). p values are indicated. Source data are provided as a Source Data file. EF, ejection fractions; FS, fractional shortening; LVIDd, left ventricular internal dimension at end-diastole; LVIDs, left ventricular internal dimension at end-systole; LW, lung weight; HW, heart weight; TL, tibial length; BW, body weight; BNP, B-type natriuretic peptide; ANP, A-type natriuretic peptide; WGA, wheat germ agglutinin.

electron microscopy (TEM) and observed that the administration of Sema restored the swelling of the mitochondrial matrix and the shortening and reduction in cristae induced by TAC (Fig. 3B, C). Additionally, the reduction abundance of mitochondria induced by TAC. After Sema treatment, mitochondrial numbers are restored (Fig. 3D). Since mitochondrial fragmentation is mainly caused by mitochondrial fission and fusion imbalance or mitochondrial apoptosis[29], we further verified the protein and mRNA expression levels of mitochondrial fission (Drp1), fusion (Opa1 and Mfn2), apoptosis (Tom20 and COX IV) and mitochondrial respiration (SDHB, NDUFV2 and ATP5A1) markers by western blotting and PCR. The results showed that TAC induction led to increased mitochondrial fission and apoptosis and decreased fusion with impaired respiration, while Sema treatment ameliorated this imbalance and mitochondrial respiratory dysfunction (Fig. 3E, F). These results suggest that Sema treatment can improve impaired mitochondrial respiration and mitochondrial structural disturbances in TAC-induced pathological myocardial remodeling.

## Metabolome analysis shows that Sema ameliorates myocardial energy production by reducing glycolytic processes and lipid accumulation

Based on the above results of the transcriptome sequencing (Fig. S3A−C), we next performed untargeted metabolomics analysis. We found that the mass spectrometry metabolomics and lipidomics results were consistent with the transcriptome sequencing results, as there were significant differences in lipid and metabolism-related total metabolites (Fig. S4A−D). The normal heart relies primarily on fatty acid oxidation for energy supply. In patients with pathological cardiac remodeling, cardiac lipid oxidation and the TCA cycle are inhibited, glycolysis is enhanced but glucose oxidation is unchanged or reduced, and the lack of energy supply further leads to HF[13,30]. Therefore, we refined the metabolites of glycolysis and the TCA-related pathways. We found that after eight weeks of TAC, the levels of cardiac glycolysis-related metabolites (glucose 6-P, fructose 6-P, 3-PG, PEP, and pyruvate), the pentose phosphate pathway (PPP) (glucose-1-P, 6-PG, and ribose-5-P) and lipids (stearic acid et al) were significantly increased, but the amount of pyruvate entering the TCA cycle (citrate and aconitate) was reduced (Fig. 4A). Despite the increased glucose utilization, the PPP increases and the entry of pyruvate into the TCA cycle is decreased, which does not produce enough ATP for cardiac energy supply under TAC induction. The PPP is not used for myocardial energy supply but for nucleotide metabolism and fatty acid substrate production[7,31]. Surprisingly, our results show that administration of Sema reduces glycolysis levels and promotes pyruvate into the TCA cycle, thereby increasing ATP production (Fig. 4A). We further investigated lipid metabolism in TAC-induced pathological myocardial remodeling. We found that the TAC-induced increase in free fatty acids led to myocardial lipid accumulation, while myocardial lipid accumulation was significantly decreased after Sema treatment (Fig. 4B). In addition, the results of the in vitro NRVM assay with oil red O staining also confirmed that PE stimulation can also increase myocardial lipid

accumulation (Fig. 4C). We next examined the mRNA levels of glycolytic key rate-limiting enzymes (HK2), TCA rate-limiting enzymes (IDH2 and PDH), lipid uptake and transport-related enzymes (CD36 and Slc27a1) and fatty acid β-oxidation (PDK4, ACOX1 and ACOT1) (Fig. 4D, E) and found that all were consistent with the metabolic results (Fig. 4A, B). Additionally, the same results were obtained for the western blot levels of glucose transporters (GLUT1 and GLUT4) and fatty acid transporters (CD36) (Fig. 4F, G). In conclusion, Sema treatment reduced glycolysis and promoted its substrates into the TCA cycle to increase myocardial energy supply and reduce the lipid accumulation of TAC-induced pathological myocardial remodeling.

## Sema regulates glycolipid metabolism through the downregulation of Creb5/NR4a1, downstream of PI3K/AKT, by the transcriptome

To determine how Sema ameliorates cardiac remodeling under chronic load stimulation, we evaluated transcriptomic alterations in the hearts of mice in the TAC-induced and Sema-treated groups. RNA sequencing (RNA-seq) and pathway analysis of differentially expressed genes (DEGs) in left ventricular tissue revealed a strong correlation between the PI3K/AKT signaling pathway and Sema treatment (Fig. 5A). Creb5 is a crucial component of the downstream signaling pathway of PI3K/AKT[32]. It is a member of the Creb (cAMP-responsive element-binding protein) family of proteins[33]. Studies have demonstrated that the Creb family plays a regulatory role in modulating the activity of NR4a1[34]. We further validated the analysis of PI3K/AKT pathway-related DEGs using RNA-seq data and found that Sema significantly reduced TAC-induced activation of the transcription factors (Creb5 and NR4a1) (Fig. 5B). We concluded that Creb5 may act as a downstream of PI3K/Akt to regulate NR4a1. Further analysis via western blotting showed that PI3K, phosphorylated AKT (p-AKT, Ser 473), Creb5, NR4a1 and phosphorylated NR4a1 (p-NR4a1) levels were significantly elevated by TAC, and Sema treatment counteracted these changes (Fig. 5C). The results revealed that TAC induced a synchronized elevation of NR4a1 in the nucleus and mitochondria (Fig. 5D). However, the ratio of nuclear to mitochondrial NR4a1 was decreased (Fig. 5E). Sema application increased the ratio of nuclear to mitochondrial NR4a1, suggesting that Sema reduced NR4a1 expression while decreasing its translocation to mitochondria. Immunoprecipitation experiments also determined that Creb5 and NR4a1 bind to each other in the heart (Fig. 5F). NR4a1 has been extensively studied as a nuclear receptor[35]. We found that treatment with Sema restored TAC-induced elevations in NR4a1 expression and reduced nuclear-mitochondrial translocation (Fig. 5B−E, G, H). These results suggest that Sema can counteract the TAC-induced activation of the PI3K/AKT/Creb5/NR4a1 signaling pathway and reduce NR4a1 nuclear-mitochondrial translocation to ameliorate TAC-induced cardiac remodeling.

## NR4a1 knockdown ameliorates glucolipid metabolic disorders, mitochondrial dysfunction and pathological cardiac remodeling

We further verified whether Sema affects Creb5/NR4a1 to improve TAC-induced pathological cardiac remodeling and metabolic

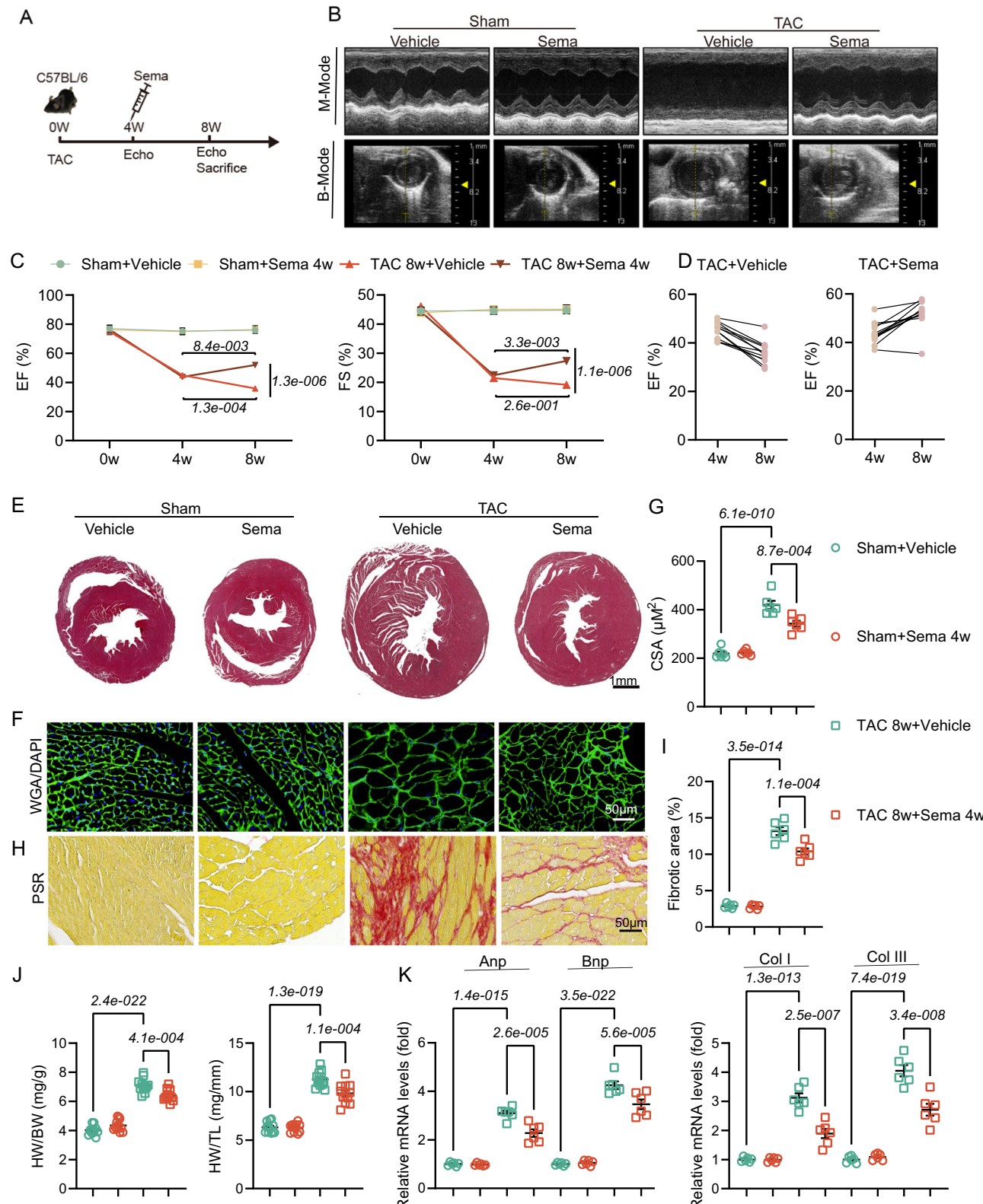

dysfunction. First, western blotting experiments showed that AKT inhibitors counteracted the elevated expression of Creb5/NR4a1 protein after TAC (Fig. S5A, B). Next, NR4a1 knockdown mice were established using AAV9-shNR4a1 to assess its role in pathological cardiac remodeling. Western blotting experiments were performed to verify the successful establishment of shRNA/shNR4a1 mice (Fig. S6A). The shRNA and shNR4a1 mice were administered Sema or vector intervention under TAC induction or sham surgery for eight weeks.

Echocardiographic results showed that NR4a1 deficiency attenuated TAC-induced cardiac dysfunction and increased both EF and FS (Fig. 6A, B). TAC-induced chronic stress resulted in a substantial increase in CSA, whereas these parameters were significantly reduced in NR4a1 knockdown mice (Fig. 6C, D). Previous studies have examined the effects of NR4a1 on mitochondrial function and structure[36]. Western blotting results suggested that NR4a1 knockdown can correct the impaired myocardial mitochondrial respiratory function caused by

**Fig. 2 | Sema reverses cardiac hypertrophy, fibrosis and dysfunction in mice that underwent TAC for eight weeks. A** The time of TAC and Sema treatment (i.p.) in mice. **B** Representative B- and M-mode echocardiographic imaging of the heart. **C, D** Echocardiographic analysis of EF and FS in mice, data are presented as mean ± SEM ($n = 6$); EF: $F_{(6, 88)} = 25.78$, $P = 1.8\text{e-}017$, $F_{(1.947, 85.69)} = 75.90$ $P = 2.0\text{e-}019$, $F_{(3, 44)} = 124.5$, $P = 1.6\text{e-}021$, $F_{(44, 88)} = 0.8118$, $P = 7.8\text{e-}001$, $F_{(6, 132)} = 27.50$, $P = 4.1\text{e-}021$, $F_{(2, 132)} = 80.98$, $P = 1.1\text{e-}023$, $F_{(3, 132)} = 107.9$, $P = 2.4\text{e-}035$; FS: $F_{(6, 132)} = 58.51$, $P = 7.9\text{e-}035$, $F_{(2, 132)} = 144.4$, $P = 5.9\text{e-}034$, $F_{(3, 132)} = 192.9$, $P = 4.6\text{e-}048$, $F_{(6, 132)} = 58.51$, $P = 7.9\text{e-}035$, $F_{(2, 132)} = 144.4$, $P = 5.9\text{e-}034$, $F_{(3, 132)} = 192.9$, $P = 4.6\text{e-}048$. **E** Representative histo-pathological cross-sectional images of mice hearts (scale bar: 1 mm; $n = 6$). **F, G** Myocardial hypertrophy was detected by WGA staining and then quantitatively analyzed (scale bar: 50 μm; $n = 6$); CSA: $F_{(3, 20)} = 70.58$, $P = 8.1\text{e-}011$.

**H, I** Myocardial fibrosis was detected by PSR staining and quantification of the collagen volume (scale bar: 50 μm; $n = 6$); Fibrotic area: $F_{(3, 20)} = 225.5$, $P = 1.4\text{e-}015$. **J** The ratio of BW, HW, and TL ($n = 12$); HW/BW: $F_{(3, 44)} = 180.2$, $P = 1.0\text{e-}024$; HW/TL: $F_{(3, 44)} = 140.2$, $P = 1.5\text{e-}022$. **K** The mRNA expression levels of ANP, BNP, collagen I and collagen III in the heart ($n = 6$); ANP and BNP: $F_{(7, 40)} = 138.8$, $P = 5.2\text{e-}026$; COLI and COL III: $F_{(7, 40)} = 82.22$, $P = 1.0\text{e-}021$. All results are shown as the mean ± SEM and analyzed using one-way ANOVA followed by Bonferroni post hoc test **G, I** and **J, K**. For the analysis in (**C,D**), repeated measures two-way ANOVA followed by Sidak post hoc test was conducted. p values are indicated. Source data are provided as a Source Data file. EF, ejection fractions; FS, fractional shortening; LW, lung weight; HW, heart weight; TL, tibial length; BW, body weight; BNP, B-type natriuretic peptide; ANP, A-type natriuretic peptide; WGA, wheat germ agglutinin; PSR, picrosirius red.

TAC-induced chronic stress (Fig. 6F–F), as evidence ATP5a, UQCRC2, MTCO1, SDHB and NDUF88 expression were significantly upregulated after NR4a1 knockdown. Impairment of mitochondrial function and structure is closely associated with HF, while disruption of glucolipid metabolism and mitochondrial damage can influence each other[37]. Then, we performed mass spectrometric metabolomic analysis of cardiac tissue from NR4a1 knockdown mice. The analysis revealed that NR4a1 knockdown counteracted the increase in glycolytic products and decreased TCA products and free fatty acid accumulation in pathological cardiac remodeling (Fig. 6G). This suggests that NR4a1 knockdown alters myocardial energy substrate utilization in response to the energy demands of pathological cardiac remodeling. Overall, these results suggest that the ameliorative effects of Sema on TAC-induced pathological cardiac remodeling are mainly dependent on the regulation of NR4a1 by Sema.

### Knockdown of NR4a1 abrogates cardiomyocyte hypertrophy and mitochondrial functional structure in vitro

To determine whether NR4a1 resists hypertrophy and ameliorates mitochondrial structure and function in isolated cardiomyocytes. We performed in vitro experiments with NRVMs and used PE to induce stress load-induced cardiac hypertrophy. We transfected NRVMs with NR4a1 siRNA or control siRNA, and the western blotting results showed successful transfection (Fig. S7A, B). The immunofluorescence results showed that the control siRNA administered with PE stimulation increased cardiomyocyte size, and the mRNA levels of BNP, ANP, and β-MHC were significantly increased. These effects were disrupted by NR4a1 knockdown (Fig. 7A, B). Then, we verified the effect of NR4a1 on mitochondria in isolated hypertrophic cardiomyocytes, and mitochondrial immunofluorescence staining showed that NR4a1 knockdown abrogated PE-induced mitochondrial division (Fig. 7C). In addition, we measured the OCR of NRVMs in vitro. In the NRVMs stimulated with PE and transfected with control siRNA, oxidative respiration was significantly decreased, and ATP production decreased. However, in NRVMs transfected with NR4a1 siRNA, oxidative respiration was significantly improved, and ATP production was increased (Fig. 7D, E). These results suggest that the knockdown of NR4a1 also abrogated PE-stimulated cardiomyocyte hypertrophy and mitochondrial dysfunction in vitro.

### NR4a1 overexpression counteracts the ameliorative effect of Sema on pathological myocardial remodeling

To provide more convincing evidence that NR4a1 in cardiomyocytes is the target for the effect of Sema on TAC-induced pathological cardiac remodeling, we established a mouse model injected with AAV9-NR4a1. We performed TAC surgery in NR4a1-overexpressed mice, and evaluated the viral efficiency in hearts 8 weeks later (Fig. S6B), the results indicated that the heart transfection was successful, but did not significantly affect the expression of NR4a1 in the liver, kidney and brain (Fig. S6B, Fig. S6E). As expected, overexpression of NR4a1 in cardiomyocytes did not exacerbate the TAC-induced reduction in LVEF and

LVFS but counteracted the therapeutic effect of Sema on TAC-induced cardiac dysfunction (Fig. 8A, B). After TAC surgery, the myocardial size and CSA of mice treated with AAV9-NR4a1 injection were also not reduced after Sema treatment (Fig. 8C–E). Furthermore, we detected the protein levels of mitochondrial respiratory chain complexes in cardiac tissues from mice with NR4a1 overexpression and those with NR4a1 knockdown and found that mitochondrial respiratory function improved in Sema-treated mice with or without NR4a1 knockdown eight weeks after TAC. In contrast, mitochondrial respiration-related protein was inhibited in mice overexpressing NR4a1 with or without Sema treatment (Fig. S6C–D). Overall, these data suggest that NR4a1 overexpression alone does not exacerbate TAC-induced pathological cardiac remodeling but counteracts the therapeutic effect of Sema, further confirming that NR4a1 is a target of Sema in pressure overload-induced cardiac remodeling and dysfunction.

## Discussion

Although numerous studies have shown that cardiac energy metabolism is altered in pathological cardiac remodeling and that the resulting myocardial energy deficit can lead to further HF, there is minimal research on therapeutic targets and drugs based on this energy shift[2,38]. Sema ameliorated and reversed the cardiac dysfunction of myocardium remodeling induced by chronic stress loading and significantly reduced cardiomyocyte volume and matrix collagen deposition. Vacuolization, swelling, and cristae breaks of mitochondria were observed in pressure-overloaded hearts. These alterations trigger disorders in the mitochondrial oxidative respiratory chain and ATP production capacity. However, these disorders can be treated and prevented by Sema. These data suggest that changes in mitochondrial structure and function induce a metabolic paradigm shift. This allows more glucose to enter the glycolytic process but not be fully oxidized. This occurs in parallel with a decrease in fatty acid consumption, resulting in lipid accumulation. The role of the pyruvate-lactate axis metabolic pattern in the control of hypertrophy has been reported[39]. In contrast, we found that Sema decreases glycolysis and free fatty acid accumulation by maintaining mitochondrial function and promoting the TCA cycle to increase the energy supply. Mechanistically, we demonstrated that Sema exerts its regulatory effect on myocardial energy metabolism via the regulation of Creb5/NR4a1 in vitro and in vivo. Transcriptional activation of NR4a1 by Creb5 disturbed the balance of glucolipid metabolism and promoted glycolysis and lipid accumulation. Sema can counteract the transcriptional activation of Creb5/NR4a1 in myocardium remodeling. In summary, we conclude that Sema is a potential drug that can improve myocardial energy metabolism in pathological cardiac remodeling and thus achieve therapeutic effects.

The majority of ATP in the healthy heart is derived from mitochondrial oxidative metabolism, whereas the end-stage failing heart undergoes a drastic shift in energy metabolism. Van Bilsen et al. proposed that in the initiation of HF, myocardial fatty acid, glucose, ketone body, and branched-chain amino acid metabolism is disturbed,

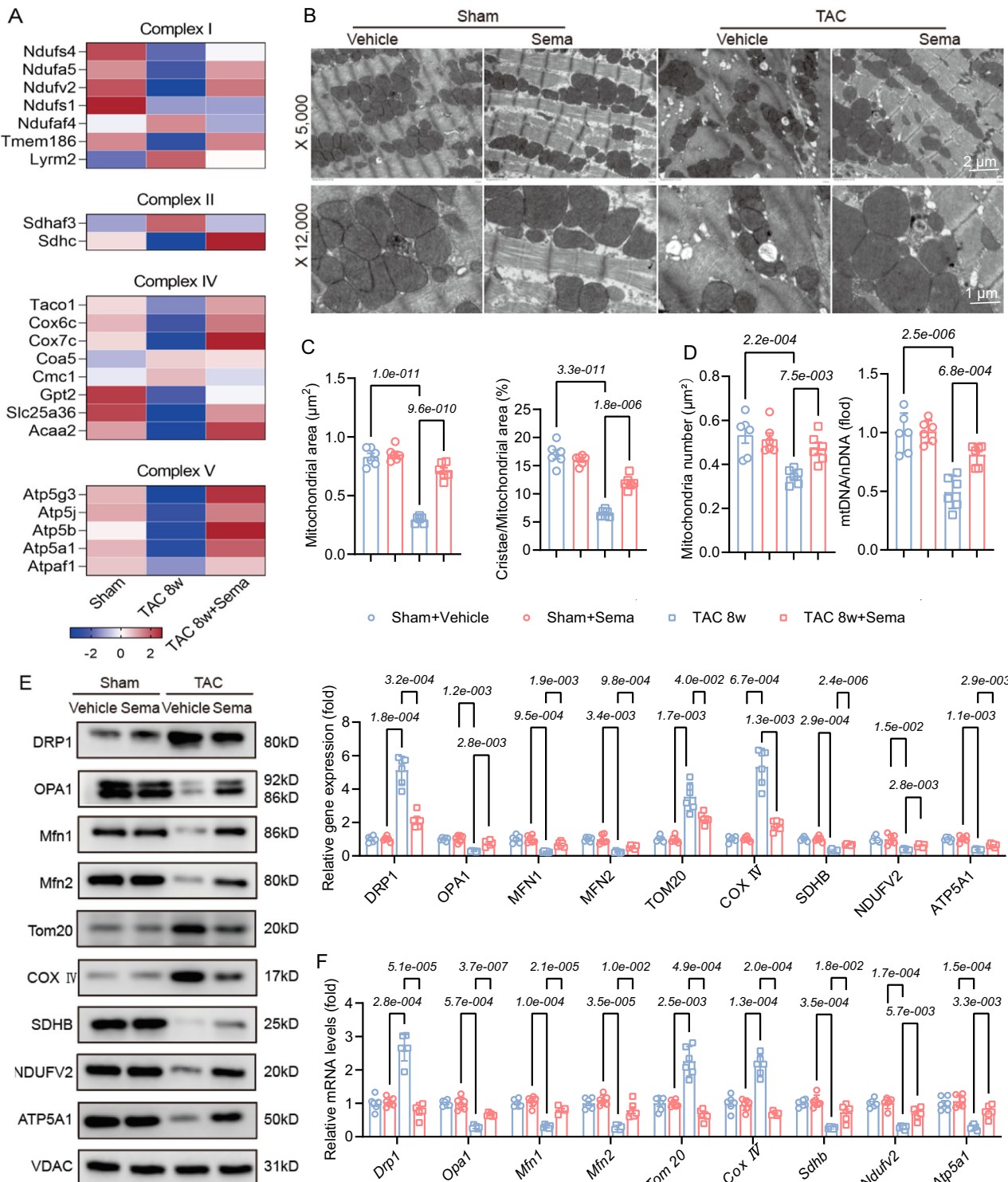

**Fig. 3 | Sema restores mitochondrial dysfunction and morphology in pathological cardiac remodeling.** **A** Heatmap showing the expression profile of mitochondrial respiratory chain signaling pathway gene sets in mouse cardiac tissue ($n = 3$). **B** TEM images of mouse cardiac tissues and quantification of the mitochondrial sectional area (scale bar: 2μm/1μm; $n = 6$). **C** The quantification of fragmented fused mitochondria ($n = 6$); Mitochondrial area: F (3, 20) = 109.4, $P = 1.4$e-012; Cristae/Mitochondrial area: F (3, 20) = 84.57, $P = 1.6$e-011. **D** The quantification of mitochondrial abundance by mitochondrial number and mtDNA/nDNA ($n = 6$); Mitochondrial number; F (3, 20) = 9.328, $P = 4.6$e-004; MtDNA/nDNA: F (3, 20) = 21.25, $P = 2.0$e-006. **E** Western blotting was used for protein quantitative analysis of mitochondrial function- and structure-related marker proteins Drp1, Opa1, Mfn2, Tom20, COX IV, SDHB, NDUFV2, and ATP5A1 and normalized to VDAC

($n = 6$ independent experiments with similar results); F (24, 160) = 67.22, $P = 4.5$e-071, F (3.243, 64.85) = 126.4, $P = 8.4$e-028, F (3, 20) = 72.44, $P = 6.4$e-011, F (20, 160) = 0.9693, $P = 5.0$e-001. **F** PCR was used for mRNA quantitative analysis of mitochondrial function- and structure-related marker proteins Drp1, Opa1, Mfn2, Tom20, COX IV, SDHB, NDUFV2 and ATP5A1 and normalized to VDAC ($n = 6$); F (24, 160) = 53.30, $P = 6.3$e-064, F (4.911, 98.21) = 49.04, $P = 5.4$e-025, F (3, 20) = 22.95, $P = 1.1$e-006, F (20, 160) = 1.417, $P = 1.2$e-001. All results are shown as the mean ± SEM, and analysis using one-way ANOVA followed by Bonferroni post hoc test (C-F) was conducted. p values are indicated. Source data are provided as a Source Data file. TEM, transmission electron microscopy; mtDNA, mitochondrial DNA; nDNA, nuclear DNA.

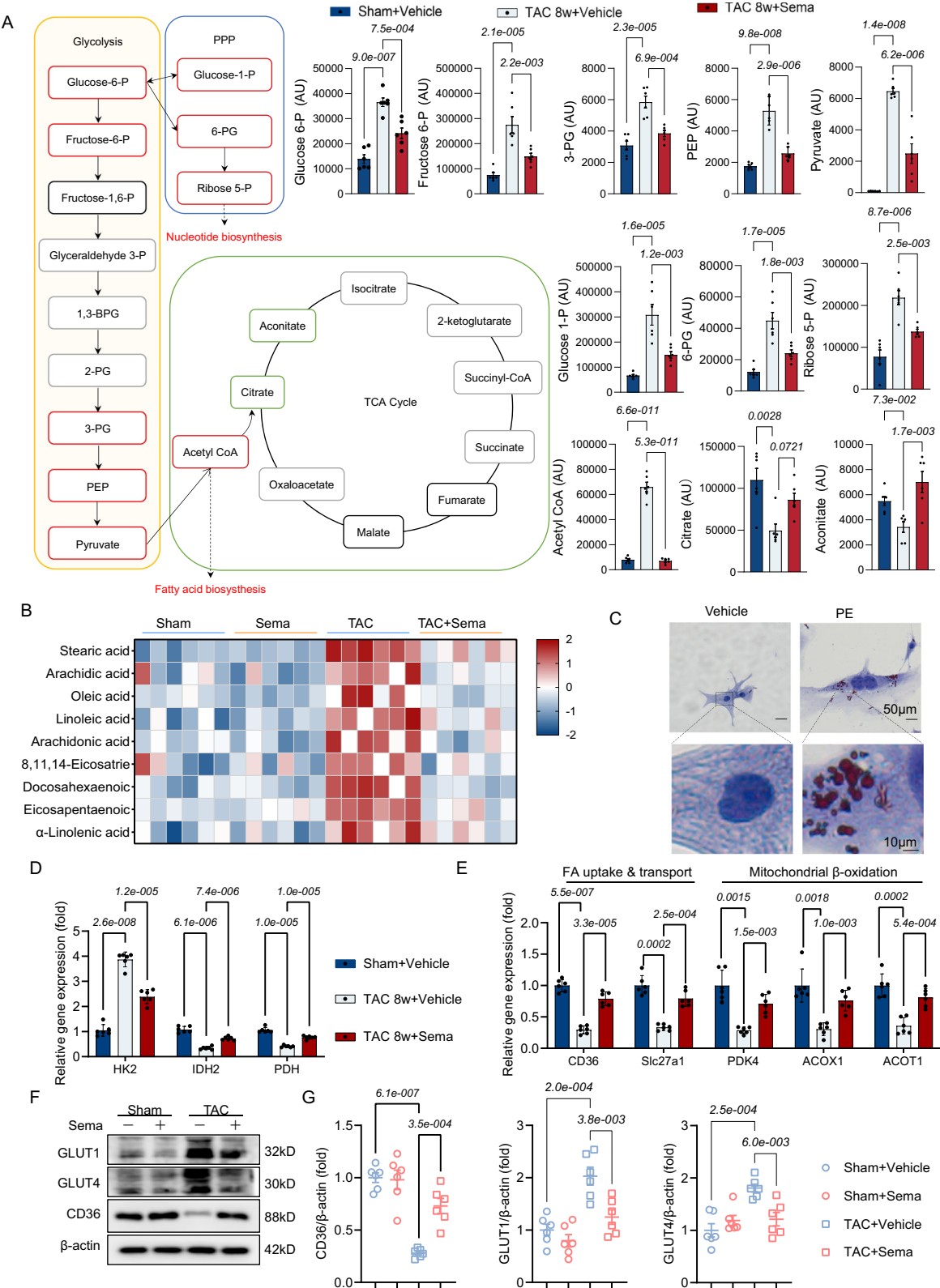

leading to alterations in myocardial structure and function[26,40]. Moreover, metabolic disturbances contribute to the disruption of mitochondrial morphology and function. Subsequently, metabolic changes and loss of mitochondrial function reinforce each other and further aggravate the process of HF[24,41]. Fusion and fission cycles are critical for mitochondrial function and biogenesis. OPA1 is obligatory for the regulation of mitochondrial inner membrane fusion, and Mfn1/2

coordinates mitochondrial outer membrane integration[42]. However, fission is executed by Drp1 in response to various physiological cues[43]. Consistently, enhanced O-GlcNAcylation of Drp1 in failing hearts promotes mitochondrial fragmentation and inhibits mitochondrial DNA repair[44]. Sema has an ameliorative effect on metabolic disorders in patients with obesity, but the mechanism is unknown. In contrast, we found that Sema regulates the expression of mitochondrial regulatory

**Fig. 4 | Sema can improve myocardial energy production by reducing glycolytic processes and lipid accumulation. A** Untargeted metabolomics mass spectrometry-based pathway analysis of Sema therapeutic cardiometabolic characteristics, quantification of changes in glycolysis and the TCA products (the red/green boxes represent products that have changed significantly; $n = 6$); Glucose 6-P: $F_{(2, 15)} = 38.02$, $P = 1.3e\text{-}006$; Fructose 6-P: $F_{(2, 15)} = 23.04$, $P = 2.7e\text{-}005$;3-PG: $F_{(2, 15)} = 23.58$, $P = 2.3e\text{-}005$; PEP: $F_{(2, 15)} = 58.41$, $P = 8.3e\text{-}008$; Pyruvate: $F_{(2, 15)} = 72.71$, $P = 1.9e\text{-}008$; Glucose 1-P: $F_{(2, 15)} = 24.41$, $P = 1.9e\text{-}005$; 6-PG: $F_{(2, 15)} = 23.85$, $P = 2.2e\text{-}005$; Ribose 5-P: $F_{(2, 15)} = 26.35$, $P = 1.2e\text{-}005$;Acetyl CoA: $F_{(2, 15)} = 206.8$ $P = 1.2e\text{-}011$; Citrate: $F_{(2, 15)} = 8.599$, $P = 0.0033$; Aconitate: $F_{(2, 15)} = 9.583$, $P = 2.1e\text{-}003$. **B** Heatmap showing the determination of the content of free fatty acids associated with untargeted lipid metabolomics ($n = 6$). **C** Oil red O staining of NRVMs (scale bar: 50μm/10μm; $n = 3$ independent experiments with similar results). **D** The mRNA expression levels of glycolysis and TCA cycle key rate-limiting enzymes HK2, IDH2 and PDH ($n = 6$); $F_{(4, 30)} = 208.2$, $P = 2.0e\text{-}021$, $F_{(1.325,}$ 19.87$) = 588.5$, $P = 3.9e\text{-}017$, $F_{(2, 15)} = 38.78$, $P = 1.2e\text{-}006$, $F_{(15, 30)} = 0.8990$, $P = 5.7e\text{-}001$. **E** The mRNA expression levels of key enzymes associated with fatty acid uptake and transport and mitochondrial β-oxidation CD36, Slc27a1, PDK4, ACOX1 and ACOT1 ($n = 6$); $F_{(8, 60)} = 0.1$, $P > 9.9e\text{-}001$, $F_{(3, 41)} = 0.3$, $P = 7.8e\text{-}001$, $F_{(2, 15)} = 426$, $P = 6.1e\text{-}014$, $F_{(15, 60)} = 0.3$, $P = 9.9e\text{-}001$. **F** Western blot analysis of the glucose transport-associated proteins GLUT1 and GLUT4 and the lipid-associated transporter CD36 ($n = 6$ independent experiments with similar results). **G.** GLUT1, GLUT4, and CD36 protein quantification levels were normalized to β-actin ($n = 6$); CD36/β-actin: $F_{(3, 20)} = 27.98$, $P = 2.3e\text{-}007$; GLUT1/β-actin: $F_{(3, 20)} = 15.61$, $P = 1.8e\text{-}005$; GLUT4/β-actin: $F_{(3, 20)} = 10.23$, $P = 2.7e\text{-}004$. All results are shown as the mean ± SEM, and analysis using one-way ANOVA followed by Bonferroni post hoc test (**A**, **D**, **E** and **G**) was conducted. p values are indicated. Source data are provided as a Source Data file. TCA, the tricarboxylic acid; NRVMs, neonatal rat ventricular myocytes.

proteins, reduces mitochondrial fission, and restores the expression of mitochondrial respiration-related complex proteins in response to stress. Our study verified that Sema markedly improved mitochondrial morphology and function in HF (Fig. 5), which may provide a mechanism for Sema to act on metabolic disorders.

Myocardial energy substrates are biased towards glycolysis to ensure a continuous energy output. Increased glycolysis increases pyruvate production, but reduced pyruvate oxidation leads to a reduction in substrates entering the TCA to produce ATP. Excess pyruvate produces lactic acid, and high lactate levels increase visceral fibrosis and cardiac dysfunction[45]. A decrease in the expression of fatty acid transporter proteins and key rate-limiting enzymes of β-oxidation impairs fatty acid oxidation. The mismatch between uptake and oxidation results in lipid accumulation, thereby producing unfavorable lipid substances, such as ceramides and diacylglycerols. This lipid accumulation leads to mitochondrial dysfunction, which in turn exacerbates HF[41]. Fatty acid accumulation causes insulin resistance through insulin signaling cascades, which counteracts increased glucose uptake and glycolysis[46]. Glucose serves as the most efficient energy substrate (ATP production/O$_2$ consumption). Theoretically, increased glucose utilization could alleviate HF energy deficiency. Nevertheless, we found a reduction in the expression of fatty acid transport proteins (e.g., CD36) and β-oxidation key rate-limiting enzymes (e.g., ACOX1) upon pressure overload, which caused a reduction in fat oxidation and fatty acid accumulation (Fig. 4B, E). Simultaneously, mitochondrial respiratory chain-associated complexes (e.g., ATP synthase) function abnormally and deter complete glucose oxidation (Figs. 3A, 4A). Increased myocardial glycolysis leads to enhanced O-GlcNAcylation-mediated mitochondrial dysfunction and structural damage[47]. However, O-GlcNAcylation interferes with electron transport chain processes and enzymes involved in aerobic respiration, further interfering with substrate oxidation processes[33,48]. Moreover, the upregulation of glycolytic bypass pathways, such as Hexosamine Biosynthetic Pathway (HBP), exacerbates glycation levels and contributes to cardiac hypertrophy[49]. Based on the reduced fatty acid oxidation and lipid accumulation in the remodeled myocardium, can we increase fatty acid oxidation to reverse cardiac function? Surprisingly, clinical trials have shown that beta-lipid-lowering drugs do not affect new-onset HF[50]. Furthermore, the use of acipimox (a lipolytic inhibitor), which substantially reduces free fatty acids but causes a sudden decline in cardiac function in patients with dilated cardiomyopathy, has no effect on healthy individuals[51]. We can conclude that directly reversing the metabolism of the failing heart back to fatty acid oxidation as a substrate for myocardial energy may instead be detrimental to cardiac function. Therefore, directly promoting glucose oxidation, rather than reversing it to normal metabolism, may be more useful to improve energy metabolism in the failing heart.

The ongoing quest for efficacious treatments in HF has underscored the imperative need for therapeutic agents capable of enhancing mitochondrial function and promoting glucose oxidation, thereby addressing the critical energy deficit prevalent among HF patients. Despite the potential of certain compounds, such as DCA and elamipretide, to augment these metabolic pathways, their clinical application remains constrained by adverse toxicological profiles and insufficient efficacy in patient populations[10,12]. This delineates a significant therapeutic void, necessitating the exploration of compounds with minimized side effects and heightened clinical action. Further expanding our understanding of metabolic regulation post-myocardial infarction, GLP-1RA have emerged as vital players[52]. Untargeted metabolomic analyzes conducted on the ventricles of infarcted mice reveal that mitochondria and metabolism constitute the principal targets of short-acting GLP-1RA therapy in the aftermath of an infarction[53,54]. In this context, our study introduces Sema as a promising candidate poised to bridge this gap. Notably, Sema has demonstrated its potential by engaging in the regulation of the oxidative respiratory chain complex. This involves key processes such as the dimerization of ATP synthase (Complex V), a pivotal step in maintaining mitochondrial function and morphology. Such regulatory mechanisms are crucial for enhancing ATP production, which in turn meets the increased myocardial energy demands by attenuating free fatty acid concentrations and optimizing the TCA cycle.

Acetyl coenzyme A, which is increased during glycolysis, is one of the substrates of glycosylation (Fig. 4A). The glycolytic bypass HBP pathway is increased after TAC, and glycosylation levels are increased[49]. However, the increases in glycosylation levels due to HBP can increase susceptibility to HF[55]. The PPP produces lipid precursors, promotes nucleotide metabolism and increases NADPH production[56]. However, G6PD glycosylation reduces the production of NADPH, allowing elevations in the PPP that promote lipid and nucleotide synthesis[57]. Moreover, nucleotide metabolism and cardiomyocyte hypertrophy are closely related[58]. This suggests that HBP and PPP products may interact to influence cardiac hypertrophy. Our study confirms this conjecture, as the activation of the PPP was consistent with lipid accumulation and cellular hypertrophy (Figs. 1 and 4).

NR4a1 is a pivotal transcription factor that plays a significant role in pathological myocardial remodeling. Reduced expression of NR4a1 and its dependent activation of the HBP have been shown to mitigate adverse myocardial remodeling[59]. Additionally, NR4a1 has been identified as a key regulator in inhibiting cardiac fibrosis following myocardial infarction by modulating glycolysis[60]. Despite its crucial functions, NR4a1 lacks an identified endogenous ligand[61]. Studies have identified several structurally diverse compounds that activate or inactivate nuclear NR4a1[62]. Notably, Cytosporone B, a natural agonist for NR4a1, has been explored for its therapeutic potential in cancer and fibrosis-related diseases[63]. Moreover, the compound ethyl 2-[2,3,4-tri-methoxy-6-(1-octanoyl) phenyl] acetate (TMPA) has demonstrated antagonistic effects on NR4a1 function in diabetic mice, offering promising avenues for metabolic disorder management[64]. However, the

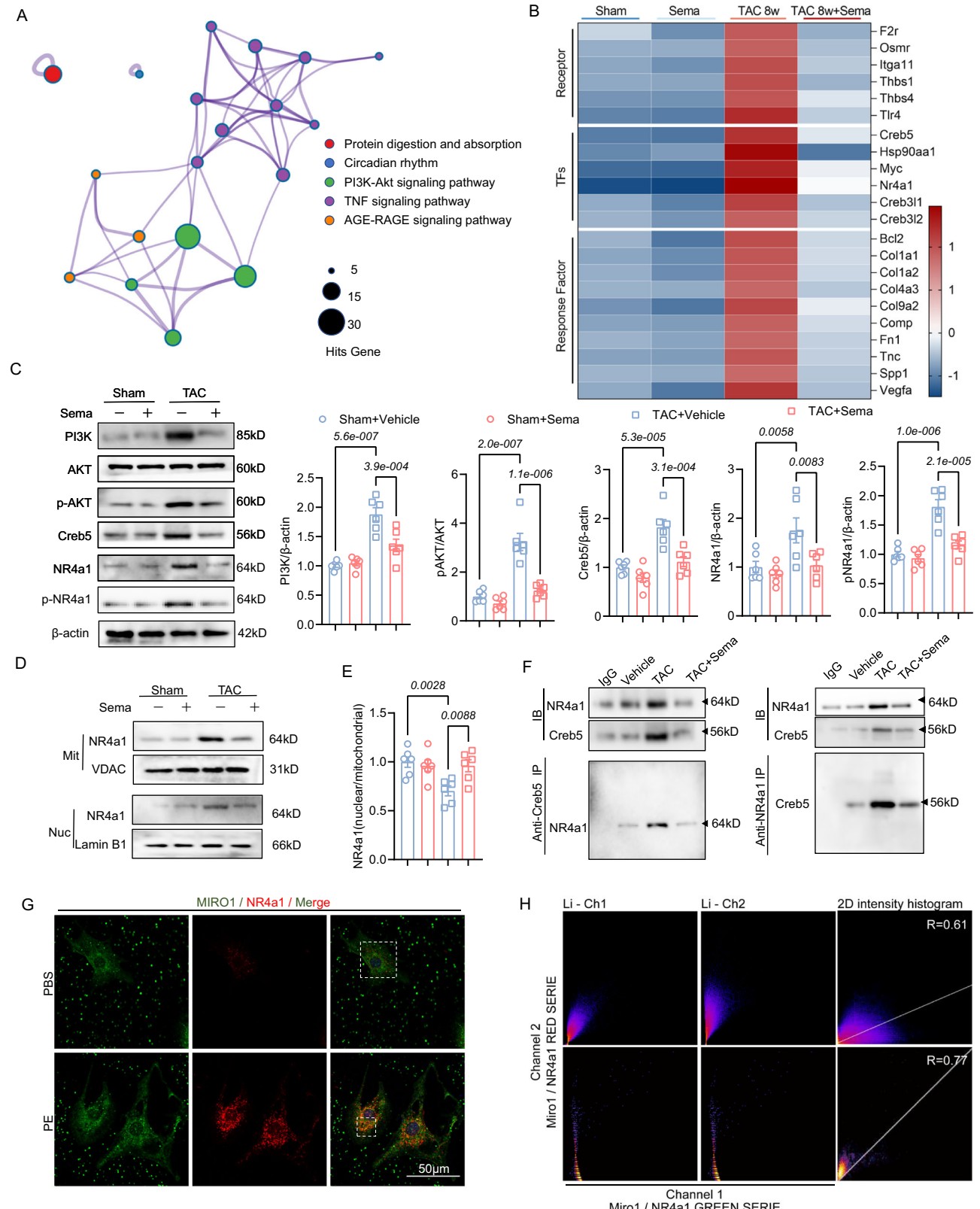

implications of these compounds for nuclear transcription modulation of NR4a1 and their effects on nuclear translocation within the context of cardiovascular disease remain unexplored. The dual nature of NR4a1's activity necessitates a balanced approach to its modulation, reflecting the variable trends observed across different experimental models. The regulation of NR4a1 extends beyond direct modulators, involving critical signaling pathways such as PI3K/AKT and

downstream factors like Creb[65]. Creb has been implicated in neuroin-flammation amelioration and blood-brain barrier stabilization post-intracerebral hemorrhage through NR4a1 regulation[34]. Given Creb5's role as an important downstream effector of PI3K/AKT and a member of the Creb family, the potential regulatory influence of Creb5 on NR4a1 warrants investigation. Our study introduces Sema, highlighting its capacity to correct abnormal NR4a1 expression mediated by Creb5

**Fig. 5 | Sema can counteract the PI3K/AKT/Creb5 signaling pathway and reduce NR4a1 expression and mitochondrial-nuclear transport. A** Based on joint DEGs and Venn diagram analysis, network maps show the key molecular pathways involved in the cardiac remodeling process and regulated by Sema ($n = 3$).
**B** Heatmap showing key molecular signatures involved in the cardiac remodeling process and regulated by Sema ($n = 3$). **C** Western blot analysis and quantification of PI3K, AKT, p-AKT, Creb5, NR4a1 and p-NR4a1 in the cardiac tissue of Sema-treated mice 8 weeks after sham or TAC surgery and quantification levels were normalized to β-actin ($n = 6$ independent experiments with similar results); PI3K/β-actin: $F_{(3, 20)} = 24.21$, $P = 7.3e\text{-}007$; pAKT/AKT: $F_{(3, 20)} = 34.29$, $P = 4.5e\text{-}008$; Creb5/β-actin: $F_{(3, 20)} = 17.38$, $P = 8.6e\text{-}006$; NR4a1/β-actin: $F_{(3, 20)} = 6.550$, $P = 0.0029$; pNR4a1/β-actin: $F_{(3, 20)} = 25.74$, $P = 4.5e\text{-}007$. **D, E** Western blot analysis and quantification of NR4a1 in mitochondrial and nuclear proteins and quantification levels were normalized NR4a1 (nuclear/mitochondrial) ($n = 6$ independent experiments with similar results); NR4a1 (nuclear/mitochondrial): $F_{(3, 20)} = 5.807$, $P = 5.0e\text{-}003$.
**F** Pull-down analysis showing the binding between Creb5 and NR4a1 in vitro ($n = 6$ independent experiments with similar results). **G, H** Representative images of MIRO1 (green) and NR4a1 (red) immunofluorescence staining in NRVMs are shown on the left, and colocalization Pearson's correlation coefficient (PCC) analysis is shown on the right (scale bar: 50 μm; $n = 3$ independent experiments with similar results). All results are shown as the mean ± SEM, and analysis using one-way ANOVA followed by Bonferroni post hoc test (**C** and **E**) was conducted. For the analysis in (**G** and **H**), a co-localization PCC analysis was used. p values are indicated. Source data are provided as a Source Data file. DEGs, differentially expressed genes; TAC, transverse aortic constriction; NRVMs, neonatal rat ventricular myocytes.

in the TAC model, thus ameliorating metabolic dysregulation associated with myocardial remodeling. Transcriptomic analyzes have further elucidated the dependency of Sema's therapeutic effects on the Creb5/NR4a1 axis, specifically in the regulation of glycolipid metabolism during pathological myocardial remodeling (Fig. 5).

Notably, NR4a1's translocation from the nucleus to mitochondria underscores its vital role in mitochondrial function regulation[66], suggesting that targeting NR4a1 to improve metabolism may offer significant therapeutic benefits. Our research demonstrated increased NR4a1 expression and phosphorylation under stress conditions, accompanied by its nuclear-to-mitochondrial translocation, thereby disrupting mitochondrial integrity (Figs. 5 and 6). Through the employment of adenoviral constructs for NR4a1 knockout and over-expression, we validated Sema's beneficial effects on glucolipid metabolism and mitochondrial dysfunction in pathological myocardial remodeling via NR4a1 modulation (Figs. 6–8). Sequencing data from the Sema reversal experiment indicated a consistent trend in genes within the Creb5/NR4a1 pathway with those observed in initial Sema treatment, highlighting the significance of this regulatory axis (Fig. S2C, Fig. 5B). Moreover, the regulation of cell migration, primarily mediated by the MAPK and ERK cascades (Fig. S2E), was validated to influence myofibroblast behavior through NR4a1 modulation[67]. Moreover, metformin's ability to ameliorate hyperglycemia-induced endothelial dysfunction by modulating NR4a1[68], suggests a close relationship between NR4a1 and vascular development. Additionally, NR4a1 is implicated in high-fat-associated endothelial dysfunction through the promotion of the CaMKII-Parkin-mitochondrial autophagy pathway in endothelial cells[69], highlighting the importance of mitochondrial pathway regulation by NR4a1. The intricate relationship between NR4a1 and various metabolic and cellular pathways underscores its therapeutic potential, particularly in myocardial remodeling and associated metabolic disorders. Our findings concerning Sema's modulation of NR4a1 through the Creb5 pathway offer insights into potential treatments for cardiovascular diseases, paving the way for future research in this domain.

The map of drug targeting at single-cell predicts the Sema showed strongest effects on P cells compared with other cardiac cells. And P cells were found to promote autonomic nervous activity through nerve growth factor (NGF) signaling[70]. Acetylcholine released by autonomic nervous of the vagal nerve in hearts mainly acts on myocardial M muscarinic receptors. In the cardioprotection provided by remote ischemic regulation, the vagus nerve stimulates intestinal peptide secretion of GLP-1, and then GLP-1R activation reduces the size of myocardial infarction by a mechanism involving M muscarinic receptors[71]. M muscarinic receptor activation has been shown to improve cardiac function in various heart diseases[72,73]. Additionally, recent research has included muscarinic receptor activation as a potential cardiac beneficial signaling pathway in the analysis of myocardial transcriptomics in mice with myocardial infarction[74]. Therefore, we speculate that the protective effect of Sema in myocardium remodeling may also be related to the activation of M muscarinic receptors. We performed M muscarinic receptor blockade experiments with Atro and found that the improvement of Sema on cardiac function were significantly blocked by Atro (Fig. S8), and Atro blockade does not affect the weight loss effect of Sema (Fig. S8A), which also suggests that the effect of Sema on the heart may be independent of its effect on body weight. Recent studies have confirmed that the anti-inflammatory benefits of Sema require GLP-1R in the central nervous system (CNS), and Atro blocks do not affect the anti-inflammatory effects of GLP-1RA[75]. This difference may be due to the fact that GLP-1R in the CNS is predominantly located on neurons and GLP-1R in the heart is predominantly located on P cells, and the physiological functions of the two types of cells are very different. In summary, we believe that Sema may exert cardioprotective effects by activating GLP-1R expressed by P cells, triggering the release of acetylcholine, and regulating ventricular function by activating M muscarinic receptors. Although this conclusion is not supported by exhaustive experiments, these findings still provide the possibility that Sema modulates ventricular muscle through acetylcholine.

In conclusion, as GLP-1RA glucose-lowering drugs have been increasingly used in clinical practice, their cardiovascular benefits have also been identified. Our study validated the clinically relevant finding that Sema, a GLP-1RA, has significant therapeutic effects on pathological myocardial remodeling. More importantly, Sema was shown to regulate myocardial energy metabolism based on the regulation of NR4a1 in pathological cardiac remodeling. Specifically, Sema not only reduces glycolysis and free fatty acid accumulation but also increases glucose oxidation and promotes the conversion of its metabolite pyruvate to the TCA cycle. Thus, the TCA produces more ATP for myocardial energy supply, which ameliorates hypertrophy, fibrosis and cardiac dysfunction in pathological cardiac remodeling. In conclusion, this study not only identifies Sema as a potential therapeutic agent for pathological myocardial remodeling but also suggests NR4a1 as a key target for regulating the myocardial glucolipid metabolic pathway. More importantly, the study may have valuable implications for patients with HF with obesity in clinical practice.

## Limitations of the study
In this comprehensive evaluation of Sema's impact on myocardial remodeling, our investigation employed a dose-response approach, administering 4.0, 12.0, or 60.0 mg/kg/day of Sema. The findings highlight a dose-dependent reduction in body weight across these doses without liver function compromise (Fig. 1), with both the lowest and highest doses of Sema providing protective effects against pathological myocardial remodeling. Interestingly, the intermediate dose of 12.0 mg/kg/day was associated with diminished cardiac function, suggesting a nuanced relationship between Sema dosage and cardiac health that warrants further exploration to understand the underlying mechanisms fully. Moreover, the exclusive use of male mice limits the generalizability of these findings across genders, pointing to the need for gender-inclusive studies to fully ascertain the therapeutic potential of Sema. Furthermore, while the TAC model is a robust method for simulating pressure overload-induced cardiac

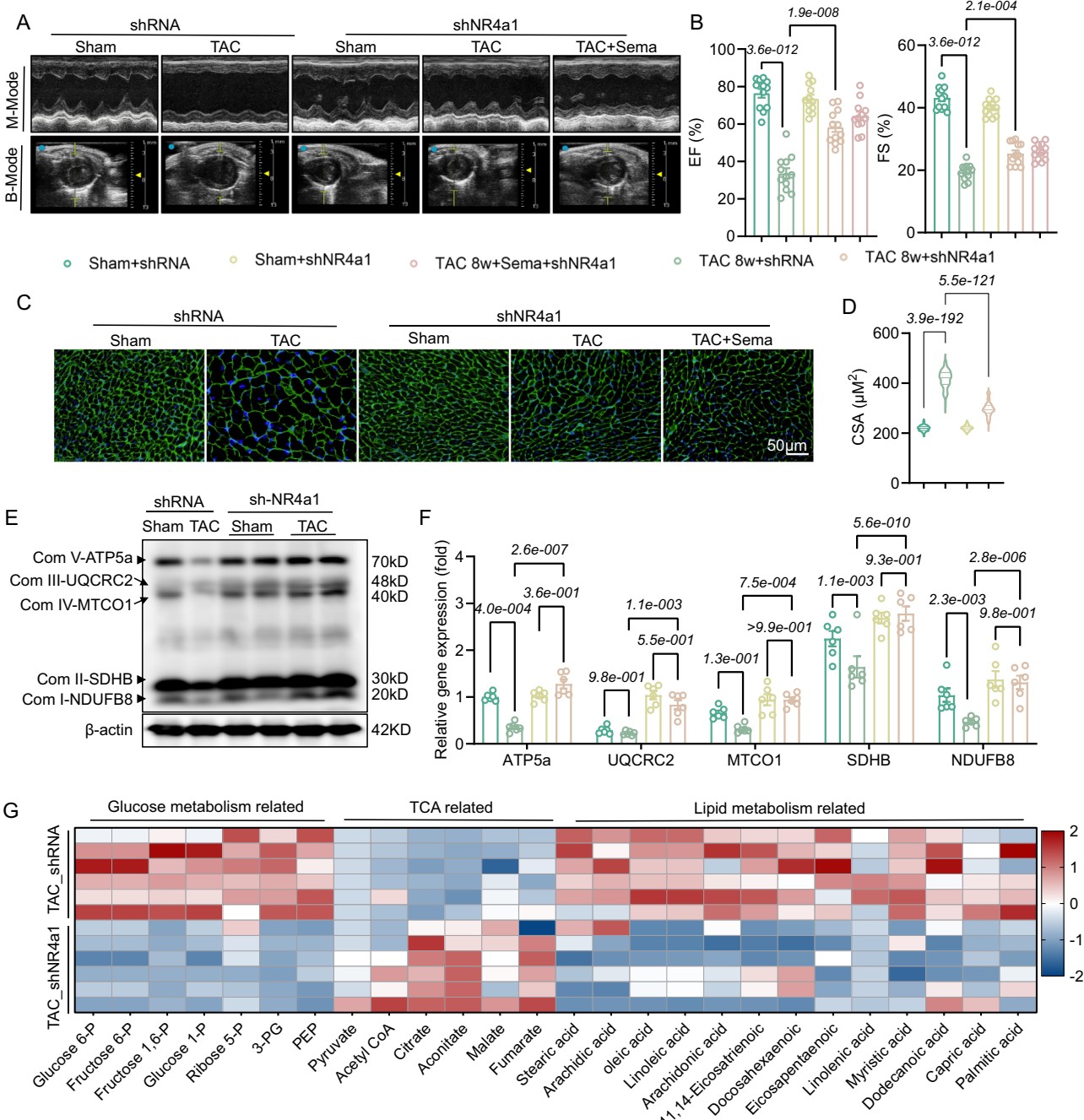

**Fig. 6 | NR4a1 knockdown ameliorates disorders of glucolipid metabolism and ameliorates pathological cardiac remodeling. A** Representative B- and M-mode echocardiographic imaging of shRNA/shNR4a1 mouse hearts. **B** Echocardiographic analysis of EF and ES in mice (n = 12); EF: F (4, 55) = 48.91, P = 1.7e-017; FS: F (4, 55) = 130.9, P = 2.0e-027. **C, D** Myocardial hypertrophy was detected by WGA staining and then quantitatively analyzed in shRNA/shNR4a1 mice (n = 6); F (3, 396) = 2018, P = 1.70e-239. **E, F** Western blot analysis and quantification of the mitochondrial respiratory chain proteins ATP5a, UQCRC2, MTCO1, SDHB and NDUF88 in the cardiac tissue of shRNA/shNR4a1 mice after sham or TAC surgery

(n = 6 independent experiments with similar results); F (12, 100) = 1.789, P = 6.0e-002, F (4, 100) = 161.5, P = 1.0e-042, F (3, 100) = 62.10, P = 9.4e-023. **G** Mass spectrometry-based analysis of glycolipid metabolism and TCA-related substances in shRNA/shNR4a1 mice 8 weeks after sham or TAC surgery (scale bar: 50 μm; n = 6). All results are shown as the mean ± SEM, and analysis using one-way ANOVA followed by Bonferroni post hoc test (**B**, **D** and **F**) was conducted. p values are indicated. Source data are provided as a Source Data file. EF, ejection fractions; FS, fractional shortening; WGA, wheat germ agglutinin; TAC, transverse aortic constriction.

hypertrophy in rodents[76], it falls short of capturing the multifaceted etiology of human cardiac disease, including genetic predispositions and susceptibility factors. This limitation emphasizes the importance of employing diverse experimental models to enhance our understanding of Sema's therapeutic implications in a broader, more clinically relevant context.

## Methods

### Animals

All animal experiments were performed using male mice to ensure that sex difference influences were excluded, and all animals were housed in specific pathogen-free (SPF) facilities condition with 20-25 °C temperature and 45-55% humidity on a regular 12-hour lightdark cycle, and

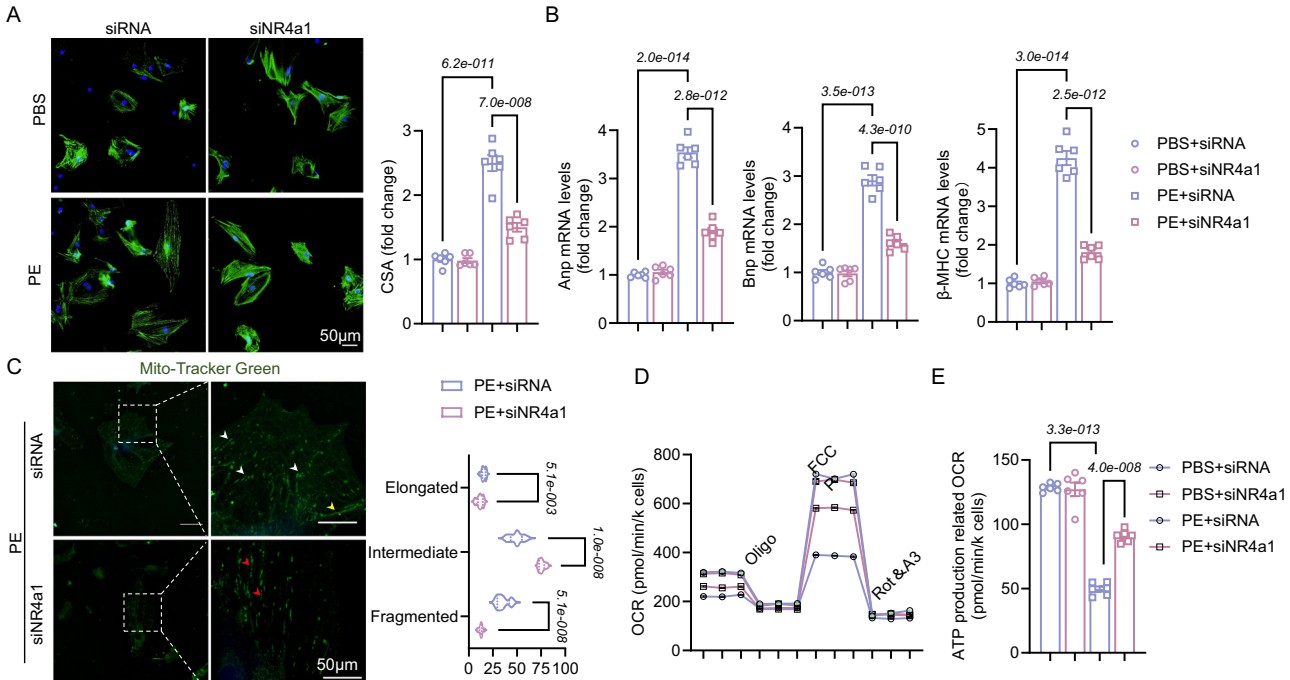

**Fig. 7 | NR4a1 knockdown improves the effects of pathological cardiac remodeling and mitochondrial function in vitro. A** Immunofluorescence staining and quantification of a-actinin in NRVMs reproducing siRNA/siNR4a1 virus after PE/PBS stimulation (scale bar: 50 μm; $n = 3$ independent experiments with similar results); $F_{(3, 20)} = 86.03$, $P = 1.3e\text{-}011$. **B** The mRNA levels of the myocardial hypertrophy indicators ANP, BNP and β-MHC (scale bar: 50 μm; $n = 6$); Anp: $F_{(3, 20)} = 279.7$, $P = 1.7e\text{-}016$; Bnp: $F_{(3, 20)} = 149.5$, $P = 7.3e\text{-}014$; β-MHC: $F_{(3, 20)} = 213.2$, $P = 2.4e\text{-}015$. **C** Representative confocal image of mitochondrial morphology stained by MitoTracker and the quantification of fragmented, intermediate, and elongated mitochondria (scale bar: 50/20 μm; $n = 6$). **D** NRVMs transfected with siRNA and siNR4a1 were subjected to an OCR assay. OCR was normalized by the total number of cardiomyocytes in each group (six pore cells per group; $n = 6$). **E** ATP production-coupled respiration in D ($n = 6$); $F_{(3, 20)} = 149.7$, $P = 7.2e\text{-}014$. All results are shown as the mean ± SEM, and analysis using a one-way ANOVA followed by Bonferroni post hoc test (**A–C** and **E**) was conducted. For the analysis in **D**, repeated measures two-way ANOVA followed by Sidak post hoc test was conducted. p values are indicated. Source data are provided as a Source Data file. PE, phenylephrine; NRVMs, neonatal rat ventricular myocytes; OCR, oxygen consumption rate; FCCP, carbonyl cyanide 4-(trifluoromethoxy) phenylhydrazone.

approved by the Animal Welfare Ethics Committee at Renmin Hospital of Wuhan University (No. WDRM20220803B). All mice were fed with an irradiated chow diet (#1035 for reproductive feeding and #1025 for maintenance feeding, Beijing HFK Bioscience Co., Ltd, Beijing, China), with free access to drinking water. C57BL/6 J male mice were purchased from the Institute of Laboratory Animal Science, Chinese Academy of Medical Sciences (Beijing, China). The 8-week-old male mice were acclimated in a quarantine room for 1 week before the subsequent experiments. In the TAC surgical induce model of pathological myocardial remodeling under chronic pressure overload, all the male mice injected with Sema and its vehicle underwent TAC surgery or sham surgery as a control group[77]. Sema was purchased from the Master of Bioactive Molecules, and was used based on available studies and biopotency conversions[78,79]. Mice were injected intraperitoneally with doses of 4.0, 12.0 or 60.0 μg/kg/day or the control drug-solution vehicle for eight weeks for subsequent experiments[79]. Atropine (Atro) was purchased from the Master of Bioactive Molecules. The male mice were intraperitoneally injected with Atro 1 mg/kg/day for blocking M muscarinic receptors. Body weight was monitored and recorded weekly.

## Adeno-associated virus 9 construction of NR4a1 overexpression or knockdown mouse model

To specifically knock down NR4a1 in cardiomyocytes, adeno-associated virus 9 (AAV9) carrying short hairpin NR4a1 (AAV9-shNR4a1) with cTNT as the promoter and its control vector (AAV9-shRNA) were constructed by Design Gene Biotechnology (Shanghai, China). The AAV9-shNR4a1 or AAV9-shRNA was orbitally injected at 50 μl per mouse ($1 \times 10^{13}$ pfu/ml). To specifically overexpress NR4a1 in cardiomyocytes, AAV9 carrying NR4a1 (AAV9-NR4a1) genomic particles and vector with green fluorescent protein (AAV9-NC) were injected in male mice via orbital veins. One week later, TAC surgery was performed and Sema treatment was administered. Then, the heart was collected eight weeks after the administration of the treatment.

## Echocardiograph

Echocardiography was performed at 4 or 8 weeks after TAC. Echocardiography was analyzed with a 10-MHz linear array ultrasound transducer equipped with a 30-MHz probe (Vevo 3100 system Visual Sonics)[77,80]. First, the mice were anaesthetised with a mixture of 4% isoflurane and 0.5 L/min oxygen. Conductive adhesive was applied to the copper plate of the physiological information monitoring table and the paws of the mice were secured to obtain electrocardiogram (ECG) and respiratory information. The probe was positioned in the parasternal short-axis view and M-mode ultrasound images were acquired for left ventricular internal dimension at end-diastole and end-systole (LVIDd and LVIDs). Multiple cardiac cycles were tracked and traced along the endocardial (Endo) trajectory of the anterior wall of the LV, and then the Endo of the posterior wall of the LV was similarly traced. On this basis, the following calculated values can be obtained: LV end-diastolic volume (EDV), LV end-systolic volume (ESV), stroke volume, ejection fractions (EF) and fractional shortening (FS). Notably, the LVFS was calculated by the formula: LVFS= (LVIDd−LVIDs)×100/LVIDd. The LVEF was also evaluated using the Teichholtz formula: LVEF = ([100 − Y]×0.15) + Y; Y = (LVIDd² − LVIDs²)×100/LVIDd².

The maximum rate of isovolumetric systolic LV pressure increase (+ dp/dt max) and maximum rate of isovolumetric diastolic LV pressure decrease (-dp/dt max) were also counted. For subsequent data

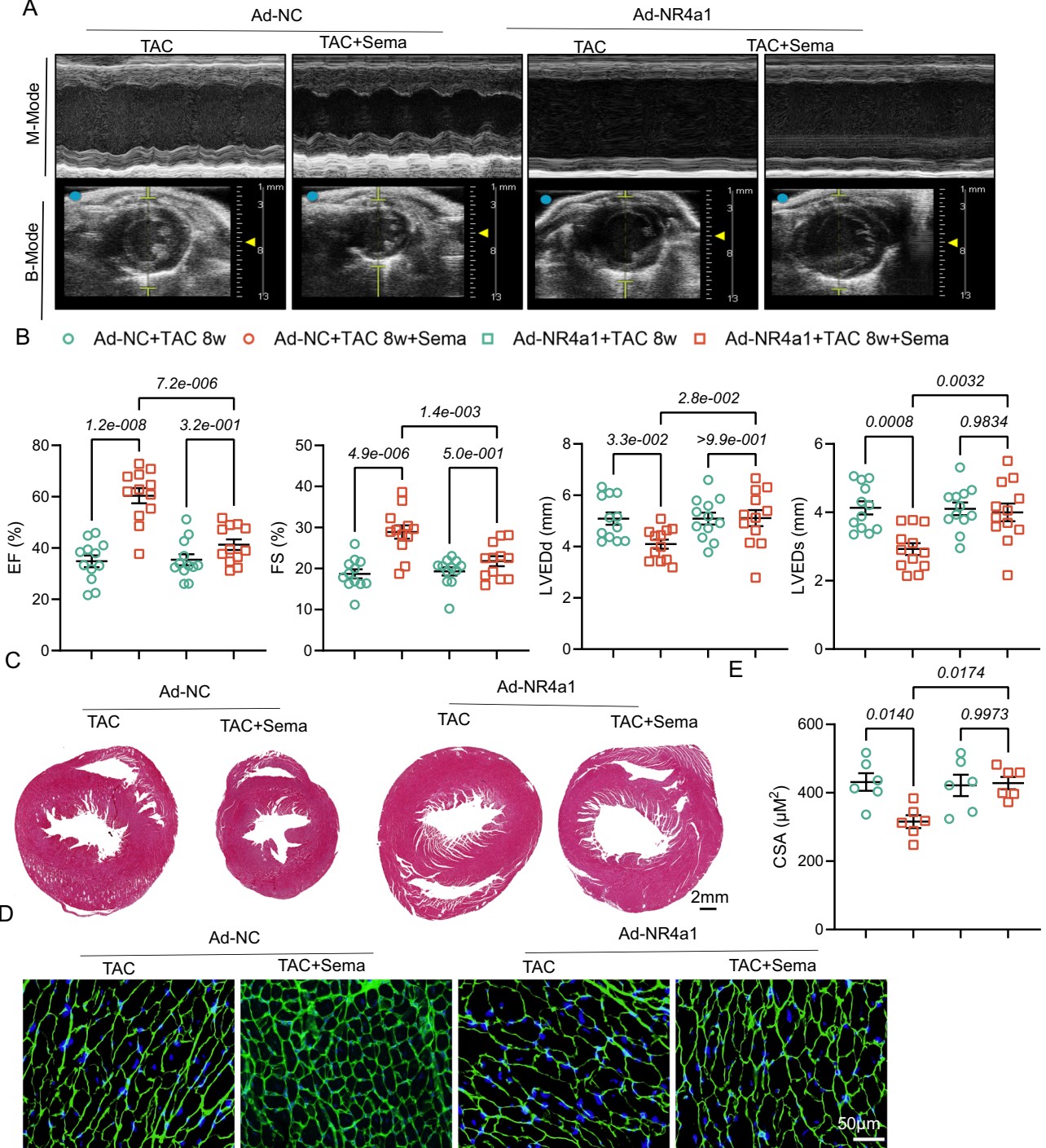

**Fig. 8 | NR4a1 overexpression exacerbates cardiac remodeling and dysfunction in mice that underwent TAC for eight weeks. A** Representative B- and M-mode ultrasound images of AAV9-NC and AAV9-NR4a1 mice 8 weeks after sham or TAC surgery. **B** Echocardiographic analysis of EF, FS, LVIDd and LVIDs in mice (*n* = 12); EF: F (3, 44) = 24.81, *P* = 1.5e-009; FS: F (3, 44) = 13.84, *P* = 1.7e-006; LVIDd: F (3, 44) = 4.115, *P* = 1.2e-002; LVIDs: F (3, 44) = 7.995, *P* = 0.0002. **C** Representative histopathological cross-sectional images of mouse hearts (scale bar: 2 mm; *n* = 6). **D**, **E** Myocardial hypertrophy was detected by WGA staining and then quantitatively analyzed (scale bar: 50 μm; *n* = 6); F (3, 20) = 5.364, *P* = 0.0071. All results are shown as the mean ± SEM, and analysis using one-way ANOVA followed by Bonferroni post hoc test (**B** and **E**) was conducted. p values are indicated. Source data are provided as a Source Data file. AAV9, adeno-associated virus 9; TAC, transverse aortic constriction; EF, ejection fractions; FS, fractional shortening; LVIDd, diastolic left ventricular internal diameters; LVIDs, systolic left ventricular internal diameters; WGA, wheat germ agglutinin.

statistics, three to five consecutive cardiac cycles were recorded continuously in the presence of nonrespiratory spikes. After the echocardiogram, mice were euthanized under anesthesia (pentobarbital sodium 80 mg/kg; intraperitoneal). Peripheral blood was collected, hearts and lungs were weighed, and tibial length was measured. The intact hearts were immersed in 10% potassium chloride and arrested in diastole. Then, the hearts were soaked in formalin for subsequent pathology-related experiments in kerosene sections. Left ventricular tissue from the mice was harvested for subsequent fractionation, transcriptional sequencing and metabolic mass spectrometry.

## Neonatal rat ventricular myocyte isolation and culture

Neonatal rat ventricular cardiomyocytes (NRVMs) were isolated from the hearts of three-day-old neonatal suckling rats[77]. In brief, the neonatal rats were euthanised with isoflurane and then the hearts were harvested. Hearts were treated 3-4 times at 37 °C with fresh $Ca^{2+}$-free HBSS solution containing 0.125% trypsin to isolate cardiac tissue. The supernatant containing the isolated cells was collected and centrifuged at 1200 rpm for 8 minutes. The supernatant is gently aspirated and medium is added to mix the cells. The isolated cells were cultured for 1 hour in DMEM containing 1.0 g/L D-glucose, 10% FBS, 100 U/mL penicillin and 0.1 mg/mL streptomycin. Cardiomyocytes suspended in the supernatant were then collected and transferred to six-well or 24-well plates and placed in a 5% $CO_2$ incubator at 37 °C for subsequent studies. Simulating the effects of in vitro drug treatment, phenylephrine (PE) stimulation was given for 48 hours, and drug treatment was given for 48 hours. Small siRNA oligonucleotides (RiboBio, Guangzhou, China) were transfected with Lipo-6000 reagent (Beyotime Biotechnology, China) according to the manufacturer's instructions. Western blotting assays were performed to demonstrate the potency of the transfected siRNA.

## Blood glucose measurements and glucose tolerance test

In the modeled mice, the blood glucose levels in both standard and diabetic animals were determined using a kit reagent (Exactech blood glucose strip) according to the glucose oxidase method. Fasting blood glucose measurements were performed weekly. After the mice were fasted, blood glucose was measured using glucose test strips (Roche, Mannheim, Germany). Briefly, the tip of the tail of the mice was cut with scissors, the first drop of blood was discarded, and the effluent blood was placed on a blood glucose test strip for blood glucose measurement. The glucose tolerance test (GTT) was performed one week before collecting peripheral blood and hearts. Specifically, mice were fasted overnight, and blood glucose levels were measured at five time points: 15, 30, 60, 90 and 120 minutes after intraperitoneal glucose injection.

## Total cholesterol, triglycerides, alanine transaminase and aspartate transaminase measurements

Orbital venous blood was collected from experimental mice and centrifuged for plasma. Enzymatic kit reagents were purchased from Nanjing Jiancheng Bioengineering Institute (Nanjing, China) and used to determine total cholesterol and triglycerides (TG). Serum concentrations of the liver enzymes alanine aminotransferase (ALT) and aspartate aminotransferase (AST) were determined by an automated biochemical analyzer (ADVIA® 2400 Siemens Ltd, Tarrytown, NY, USA)[81].

## Measuring mitochondrial respiratory function with OCR

To observe mitochondrial respiratory function, the above isolated NRVMs were cultured in cell dishes for 48 hours and placed in wells coated with 1% Matrigel for analysis with Seahorse BioSciences XF$^e$ 24 Extracellular Flux Analyzers (Seahorse BioSciences Asia, Pudong, Shanghai). After stimulating the cells with PE or Sema, measurements were taken for five consecutive days with daily medium changes, and the oxygen consumption rate (OCR) was measured using the Cell Mito Stress Kit (Seahorses Biosciences, 103015). Data were normalized to the number of cells measured on culture plates by 4-pyr6-diamino-2-phenylindole (DAPI) staining according to the Seahorse Cell Mito Stress Kit protocol with oligomycin final concentration of 1.5 μM, FCCP final concentration of 2.5 μM, and rotenone/antimycin A final concentration of 0.5 μM.

## Histological analysis and immunofluorescence staining of tissues and cells

Cardiac tissue that had been fixed in sections was stained with picrosirius red (PSR), and collagen volume was measured for analysis.

Hematoxylin-eosin staining (HE) and wheat germ agglutinin (WGA) staining were used to observe the extent of hypertrophy in the mouse myocardium in vivo. Successfully stimulated NRVMs were incubated overnight at 4 °C with primary antibodies against a-actinin (Abcam #ab108198 1:1000), NR4a1 (CST #3960 1:1000) or anti-MIRO1 (ABCAM #ab188029 1:1000)[82]. The next day, treated with secondary antibodies (goat anti-mouse IgG Alexa Fluor 488 secondary antibodies Invitrogen #A11001 IF 1:200; goat anti-rabbit IgG Alexa Fluor 568 secondary antibodies Invitrogen #A11011 IF 1:200)for 60 min at 37°C, the nuclei were stained with DAPI and sealed for photographic purposes. PSR and HE images were acquired with an Aperio VERSA 8 (Leica Biosystems), and immunofluorescence and WGA images were acquired with an Olympus BX53. All images were analyzed using Image-Pro Plus 6.0.

## Transmission electron microscopy (TEM) imaging and mitochondrial morphology assessment

TEM and mitochondrial morphological protocols were performed based on our previous studies on mitochondria[83]. After successful modeling, the mice were anesthetized and euthanized after eight weeks of TAC. The hearts were quickly removed and placed in freezing tubes in liquid nitrogen for subsequent cutting. Left ventricular cardiac tissue was cut into small pieces approximately 2-3 mm thick, immediately fixed in electron microscopic solution (2.5% glutaraldehyde) for 24 hours, stained with toluidine blue for the more conspicuous areas, stained with 4% methanolic uranyl acetate and Reynolds lead citrate stain, and observed with a transmission electron microscope to observe the ultrastructure of mouse myocardial mitochondria (TM-3000; Hitachi, Ltd, Tokyo, Japan).

## Transcriptome sequencing analysis

The total high-purity RNA from mouse cardiac tissue was first isolated and converted into raw sequences by base calling for quality control using software[84]. Then, hisat2 software was used for comparison with the reference gene group, and valid comparison reference genes were obtained by feature count soft count gene number. The basic data after using all resources and software are shown in the table (Table S1) below and used for data analysis and plotting. Differentially expressed genes between groups with Limma moderated t test correction ($P < 0.1$).

## Untargeted metabolomics analysis based on HPLC-Zeno TOF-MS/MS

In this experiment, a total of 36 samples were analyzed, consisting of six groups: TAC 8 W+Sema, sham+Sema, TAC 8 W, sham, TAC 8 W+shRNA, and TAC 8 W+shNR4a1. Each group contained six biological replicates. In addition, quality control samples were prepared by mixing nine randomly selected samples from the aforementioned 36 samples. The specific method is the mouse cardiac tissue was separated and homogenized in ultrafiltered water. Subsequently, 4 times the volume of acetonitrile-methanol solution (50% v/v) was added to the tissue homogenate, followed by ultrasonic lysis for 10 minutes and refrigeration for 1 hour to precipitate the protein. The supernatant was obtained by centrifugation and dried under high-purity nitrogen gas. The supernatant was reconstituted using 50% v/v acetonitrile in water and centrifuged again. The supernatants were then collected and transferred to vials, and quality control samples were made.

The metabolites were dissolved by liquid chromatography using mobile phase A and then separated via ExionLC™ Series UHPLC. Mobile phase A was an aqueous solution containing 0.1% formic acid; mobile phase B was acetonitrile. The liquid phase gradient settings were as follows: 0–2.0 min, 1% B; 2.0–12.0 min, 1%–99% B; 12.0–19.0 min, 99% B; 19.0–19.1 min, 99%–1% B; and 19.1–21.9 min, 1% B. The flow rate was maintained at 300 nL/min. The metabolites were separated by the Kinetex® polar C18 reverse phase column (Strata™-X,

Phenomenex, 00D-4759-AN), and then injected into an ESI ion source for ionization and then into a SCIEX-ZenoTOF 7600 system (AB Sciex Pte. Ltd). The TOF MS experiment settings were as follows: spray voltage, 5.5 kV; declustering potential, 60 V; collision energy, 10 V; and scan range, 60–1000 Da with an accumulation time of 0.15 s. The TOF MS/MS settings were as follows: declustering potential, 60 V; and collision energy, $35 \pm 15$ V. The secondary mass spectrometry scan ranged between 25 and 1000 Da. The data acquisition mode was performed using the Data Independent Acquisition (IDA) program, and the zeno threshold was set at 2000000 cps.

Metabolite identities were confirmed based on inter self-built libraries and MetDNA (http://www.metdna.zhulab.cn). The identity process settings of inter self-built libraries were as follows: 5 ppm error tolerance in precursor ion m/z and fragment ion spectra; and less than 5% shift in retention time relative to purified standard metabolites. The relative quantification of metabolites was determined by the peak area of the precursor ion, which was normalized to total protein. Statistical analysis was performed using MetaboAnalyst 5.0 (https://www.metaboanalyst.ca). Graphs were generated via R and GraphPad Prism 9.4.1 (associated data was provided in Supplementary Data 1).

### Western blotting and immunoprecipitation

To check the interaction relationship between Creb5 and NR4a1, proteins were extracted by lysis of NRVMs or cardiac tissue with a RIPA lysis buffer system (Santa Cruz Biotechnology, sc-24948). Protein concentrations of all samples were quantitatively and uniformly extracted with BCA. Then, they were configured in 10% SDS–PAGE gels, and separated by electrophoresis of equal amounts of protein on the samples. Then, the proteins were transferred to PVDF membranes with a transfer solution at a specific voltage. After the proteins were blocked in 5% skim milk for one hour, they were incubated overnight with the primary target protein antibody (anti-DRP1,SANTA #SC-32898 1:1000; anti-OPA1, SANTA #sc-30573 1:1000; anti-Mfn1, SANTA #sc-50330 1:1000; anti-Mfn2, SANTA #sc-100560 1:1000; anti-Tom20, SANTA #sc-17764 1:1000; anti-COX IV, CST #11967 1:1000; anti-SDHB, ABclonal #A23832 1:2000; anti-NDUFV2, ABclonal #A7442 1:2000; anti-ATP5A1, proteintech #14676-1-AP 1:1000; anti-VDAC, ABCAM #ab191440 1:1000; anti-GLUT1, ABclonal #A11208 1:1000; anti-GLUT4, ABCAM #ab654 1:1000; anti-CD36 proteintech #18836-1-AP 1:2000; anti-β-Actin, CST #4970 1:1000; anti-PI3K, CST #4257 1:1000; anti-Akt, CST #4691 1:1000; anti-p-Akt (Ser473), CST #4691 1:1000; anti-NR4a1, CST #3960 1:1000; anti-Creb5, ABCAM #ab168928 1:1000; anti-P-NR4a1, CST #5095 1:1000; anti-Lamin B1, proteintech #12987-1-AP 1:1000; anti-UQCRC2, ABclonal #A4366 1:1000, anti-MTCO1, ABclonal #A17889 1:1000; anti-NDUFB8, ABclonal #A19732 1:1000). The next day, the membrane was incubated with a secondary antibody of the same species (mouse or rabbit) for one hour, and immunoblotting was detected using the Chemidoc Touch Imaging System (Bio-Rad, USA). Coimmunoprecipitation assays were carried out according to the standard method purchased from Beyotime (Shanghai, China). Protein A + G agarose electrophoresis was performed according to the instructions for the experiment. Anti-immunoglobulin G antibody and anti-Creb5 antibody were incubated together with lysates of mouse left ventricular cardiac tissue overnight at 4 °C with rotation and then incubated with protein A + G agarose for 3-4 hours at 4 °C. Then, the magnetic beads were repeatedly washed 4-5 times with prechilled PBS and boiled with 1× SDS loading buffer. The supernatant was then subjected to western blotting to determine the interaction between Creb5 and NR4a1.

### Quantitative real-time polymerase chain reaction (qRT–PCR)

To determine mRNA expression, total RNA was extracted from each sample group using TRIzol reagent, and cDNA synthesis was performed using a Transcription First Strand cDNA Synthesis Kit (Roche, Basel, Switzerland). Then, 480 SYBR green dye was used for quantitative real-time PCR. Gene expression was determined by reverse transcription-polymerase chain reaction using a Roche Light-Cycler 1 detection system. RNA levels were normalized to β-actin levels, and changes in CT cycle values between groups were compared. The expression of the target genes in each group were compared using the $2^{-\Delta\Delta CT}$ method and normalized using statistical analysis (the sequence is shown in Supplementary Data 2).

### Data analysis

All data in this study are expressed as the mean ± SEM, and statistical analysis was performed using GraphPad Prism 9.4.1 software. Guided by preliminary data analysis, comparisons between two groups with a normal distribution and homogeneity of variance were performed using an unpaired two-tailed Student's t test, whereas one-way analysis of variance (ANOVA) followed by Bonferroni post hoc test was conducted for comparisons among three or more groups. Analyzes with $p < 0.05$ were considered statistically significant, and the corresponding p values are indicated above each legend. Related software and algorithms are listed in Table S1. We collected data from animal studies in a blinded manner and performed the final statistical analysis.

### Reporting summary

Further information on research design is available in the Nature Portfolio Reporting Summary linked to this article.

## Data availability

The data supporting the findings from this study are available within the manuscript and its supplementary information. Source data are provided with this paper. Any additional raw data are available from the corresponding author upon reasonable request. The datasets generated for the RNA-seq are available through the Gene Expression Omnibus under accession code GSE262105. The Metabolomic data generated in this study have been deposited in MassIVE under accession code ID: MSV000094408. Source data are provided with this paper.

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

## Acknowledgements
This work was supported by grants from The Regional Innovation and Development Joint Fund of National Natural Science Foundation of China (No. U22A20269) (QZ.T), National Key R&D Program of China (2018YFC1311300) (QZ.T), The National Natural Science Foundation of China (81530012) (QZ.T) and The Fundamental Research Funds for the Central Universities(2042023kf0016) (CY.K).

## Author contributions
QZ.T, YL.M, CY.K., and Z.G. supervised and conceptualized the study. YL.M., CY.K., and Z.G. designed and performed the majority of the experiments and analyzed the majority of the data. YL.M., CY.K., and Z.G. wrote and edited the manuscript. MY.W., P.W., D.Y.I., FY.L., and Z.Y. assisted with some experiments and discussed the results. All authors edited and approved the manuscript.

## Competing interests
The authors declare no competing interests.
