## [Peer Review File · Nature Communications]

REVIEWER COMMENTS

Reviewer #1 (Remarks to the Author):

The study described in this manuscript tested the potential role of Semaglutide (Sema) as a therapeutic in heart failure induced by pressure-overload. More specifically they treated mice with Sema following transverse aortic constriction (TAC) and performed functional, cellular, and molecular assessments. Sema is a glucagon-like peptide-1 agonist (GLP-1RA) which are often used for the treatment of diabetes. However, it is suggested that there are cardioprotective roles of these compounds, at least associated with weight loss. This study expands upon these findings to support a role of Sema in treating general heart failure. The authors show that Sema treatment can both attenuate development of cardiac remodeling and dysfunction as well as partially reverse it following TAC. Using the metabolic and molecular examination the authors define a potential mechanism for this protection. They identify changes in metabolites supporting restoration of mitochondrial function following Sema treatment and that this may be through the transcription factor Nr4a1 and downstream signaling via AKT. To this later point they authors show that Sema reduces Nr4a1 associated with the cardioprotection and direct knockdown of Nr4a1 is sufficient to replicate some of the adaptive changes in metabolic changes. Furthermore, overexpression of Nr4a1 prevented Sema-induced protection. Many of these findings were reproduced at the cellular level in vitro in response to phenylephrine (PE) treatment. Overall, the study is in an important area of research and the results support the conclusions of the authors. Only a few minor points should be addressed to further strengthen the impact of the study.

1. A few minor typos are noted. e.g., Line 83, Sem is used instead of Sema and that same sentence has some grammatical errors; Line 163, raw sequenced sequences is redundant; Line 217, states "Protein levels" in the qRT-PCR section, though I think the authors meant "RNA levels"; and others.
2. Please indicate the sex of the mice used.
3. Please provide the RNA-sequencing data in some form, e.g., submission to a data repository and/or complete table of all transcripts detected. This could also be an expansion of the data presented in Table S2 to include the individual genes instead of the pathway summaries.
4. Please indicate which phosphorylation site was measured for AKT in Figure 5, e.g., Thr308, Ser473, etc.

Reviewer #2 (Remarks to the Author):

The study investigates the impact of Semaglutide (Sema) on cardiac remodeling induced by transverse aortic constriction (TAC) in mice. Sema is found to improve cardiac dysfunction, attenuate hypertrophy and fibrosis, and maintain mitochondrial morphology and function under chronic stress conditions. Untargeted metabolomics analysis reveals positive effects on mitochondrial lesions, lipid accumulation, and ATP insufficiency. Transcriptional profiling identifies the involvement of Creb5/NR4a1 in the PI3K/AKT pathway. Sema is shown to regulate myocardial energy metabolism and protect against pathological myocardial remodeling.

Major Issues:

1. The manuscript lacks clarity in explaining the rationale behind choosing the specific doses of Sema (4 µg/kg, 12 µg/kg, and 60 µg/kg) for the experiments. The justification for selecting the high dose (60 µg/kg) should be provided.
2. The methods section does not elaborate on the criteria used for assessing blood glucose, cholesterol, and liver functions, raising concerns about the reliability of these measurements.
3. While the study demonstrates the efficacy of Sema in reversing TAC-induced cardiac dysfunction and hypertrophy, the mechanistic insights into how Sema exerts these effects are somewhat limited. Further investigations into the specific signaling pathways and molecular mechanisms involved would enhance the manuscript's scientific depth.
4. The statistical analysis section needs improvement in reporting specific statistical tests used, including justification for choosing either homogeneity of variance or heteroscedastic data.
5. The manuscript lacks a thorough discussion on potential limitations and challenges encountered during the study, such as any observed side effects of Sema or unexpected outcomes.
6. Figures and tables could be better organized and labeled for improved readability. Additionally, some figures lack detailed captions that explain the significance of the data presented.
7. The introduction could benefit from a more comprehensive literature review, highlighting existing knowledge gaps that the current study aims to address.

Minor Issues:

1. The terminology used in the abstract, such as "glycolytic species products in the tricarboxylic acid cycle (TCA)," could be clarified for better understanding.
2. The introduction contains grammatical errors and could be refined for better readability.
3. The rationale for choosing the 8-week time point for the experiments needs clarification. Was this duration based on previous studies or empirical observations?
4. The explanation of the methods for echocardiographic analyses, specifically the M-mode tracer method, requires more detail to ensure reproducibility.
5. The authors should provide more information on the potential limitations of the study, especially regarding the TAC model and its relevance to human cardiac pathology.
6. There is a need for clarification on how the study controlled for potential confounding factors in the TAC model, such as variations in surgical techniques or individual mouse responses.
7. In the discussion, the authors could provide a more comprehensive comparison of their findings with existing literature to underscore the novelty and significance of their results.

Reviewer #3 (Remarks to the Author):

In this study the authors have investigated the effects of the GLP-1R agonist semaglutide (Sema) in cardiomyocytes and a transverse aortic constriction surgery as the disease model. The authors have presented an extensive data set showing that the protective effects of Sema involve the Creb5/NR4A1 axis.

1. The authors convincingly show in the TAC model that Sema attenuates markers of hypertrophy, fibrosis and dysfunction (mitochondrial). Did the authors determine levels of ROS and effects Δ MMP in these studies?

2. The interactions of Sema, TAC, Creb5 in the disease model are of interest and need some explanation: The reference #39 which identify Creb5/NR4A1 as an important downstream pathway of P13K/Akt mentions the Akt-Creb5 connection but not NR4A1 and reference #40 is primarily focused Nur77-dependent regulation of microRNAs. Are there more relevant references that should be included? The results suggest that TAC induces nuclear export of NR4A1, and this is inhibited by Sema and may be important for TAC-induced mitochondrial damage. This should be confirmed using a nuclear export inhibitor such as leptomycin B. In addition, the changes in NR4A1 phosphorylation can be important for inducing NR4A1 nuclear export. The authors also state “Transcriptional profiling revealed that Sema exerted its effects on myocardial energy metabolism by regulating Creb5/NR4a1 in the canonical PI3K/AKT pathway. Specifically, Sema reduced the expression of NR4a1 and its translocation from the nucleus to mitochondria.” This suggests that TAC induces both extranuclear (export) and nuclear pathways (inhibited by Sema) and therefore the balance of NR4A1 (nuclear/NR4A1 (mitochondrial)) may be required for an optimal “damage” response and this needs to be addressed.

3. Do the authors have any idea how Sema, working through the GLP-1R assay affects NR4A1 expression and function and intracellular location? It should also be noted that NR4A1 ligands can also modulate receptor-dependent glucose metabolism and fibrosis and this should be integrated into the Discussion.

Response to comments

Dear Reviewers,

Re: Manuscript ID: NCOMMS-24-01828A and title: Semaglutide ameliorates cardiac remodeling by optimizing energy substrate utilization and through the Creb5/Nr4a1 axis in male mice

First and foremost, I would like to express my sincere gratitude for the time and effort you have dedicated to reviewing our manuscript. Your insightful comments and constructive suggestions are invaluable to enhancing the quality and clarity of our work. We have carefully considered each of your feedback points and have made comprehensive revisions to our manuscript accordingly. Below, we provide detailed responses to each of the specific comments and explain how we have incorporated your valuable feedback into our revised manuscript.

Thank you for your continued support.

Respond to Reviewer #1:

The study described in this manuscript tested the potential role of Semaglutide (Sema) as a therapeutic in heart failure induced by pressure-overload. More specifically they treated mice with Sema following transverse aortic constriction (TAC) and performed functional, cellular, and molecular assessments. Sema is a glucagon-like peptide-1 agonist (GLP-1RA) which are often used for the treatment of diabetes. However, it is suggested that there are cardioprotective roles of these compounds, at least associated with weight loss. This study expands upon these findings to support a role of Sema in treating general heart failure. The authors show that Sema treatment can both attenuate development of cardiac remodeling and dysfunction as well as partially reverse it following TAC. Using the metabolic and molecular examination the authors define a potential mechanism for this protection. They identify changes in metabolites supporting restoration of mitochondrial function following Sema treatment and that this may be through the transcription factor Nr4a1 and downstream signaling via AKT. To this later point they authors show that Sema reduces Nr4a1 associated with the cardioprotection and direct knockdown of Nr4a1 is sufficient to replicate some of the adaptive changes in metabolic changes. Furthermore, overexpression of Nr4a1 prevented Sema-induced protection. Many of these findings were reproduced at the cellular level in vitro in response to phenylephrine (PE) treatment. Overall, the study is in an important area of research and the results support the conclusions of the authors. Only a few minor points should be addressed to further strengthen the impact of the study.

1. A few minor typos are noted. e.g., Line 83, Sem is used instead of Sema and that same sentence has some grammatical errors; Line 163, raw sequenced sequences is redundant; Line 217, states “Protein levels” in the qRT-PCR section, though I think the authors meant “RNA levels”; and others.

Reply Q1: Thank you for pointing out the typos and grammatical errors in our manuscript. We appreciate your attention to detail, which has helped to improve the clarity and accuracy of our work. We have carefully reviewed the manuscript and corrected the errors as follows:

Line 83: We have corrected "Sem" to "**Sema**" and fixed the grammatical errors in revised manuscript.

Line 163: The corrected sentence now reads: The total high-purity RNA from mouse cardiac tissue was first isolated and **converted into raw sequences** by base calling for quality control using software.

Line 217: We have revised the sentence as follows: **RNA** levels were normalized to β -actin levels, and changes in CT cycle values between groups were compared.

We conducted a thorough review of the manuscript to identify and correct any typos or grammatical errors, ensuring the text's clarity and accuracy. Moreover, to enhance the quality of the article, we engaged a professional agency for proofreading and polishing. Below is the proofreading certificate from the agency. All these changes have been highlighted in red in the document for easy identification.

2. Please indicate the sex of the mice used.

Reply Q2: Thank you for highlighting the importance of specifying the sex of the mice used in our study. We acknowledge that the sex of an animal model can impact experimental outcomes. To this end, we exclusively utilized male mice. The manuscript has been updated to clearly indicate this choice. Notable updates include:

In the title section, the title has been changed to: “Semaglutide ameliorates cardiac remodeling by optimizing energy substrate utilization and through the Creb5/NR4a1 axis in **male** mice.”

In the Methods section:

2.1: The revised text now reads, "All animal experiments were performed using **male** mice to ensure that sex difference influences were excluded", "The 8-week-old **male** mice were acclimated in a quarantine room for 1 week before the subsequent experiments", "In the TAC surgical induce model of pathological myocardial remodeling under chronic pressure overload, all the **male** mice injected with Sema and its vehicle underwent TAC surgery or sham surgery as a control group", "The **male** mice were intraperitoneally injected with Atro 1 mg/kg/day for blocking M muscarinic receptors. Body weight was monitored and recorded weekly".

2.2: The revised text now reads, "To specifically overexpress NR4a1 in cardiomyocytes, AAV9 carrying NR4a1 (AAV9-NR4a1) genomic particles and vector with green fluorescent protein (AAV9-NC) were injected in **male** mice via orbital veins".

We appreciate the opportunity to clarify this aspect of our study and believe that these additions will make a significant contribution to the transparency and rigor of our research.

3. Please provide the RNA-sequencing data in some form, e.g., submission to a data repository and/or complete table of all transcripts detected. This could also be an expansion of the data presented in Table S2 to include the individual genes instead of the pathway summaries.

Reply Q3: Thank you for highlighting the importance of making our RNA sequencing data accessible to reviewers and the wider research community. In response to your suggestions, we have taken the following steps:

1. **Database submission:** We have submitted the complete RNA sequencing data to NCBI's Gene Expression Omnibus (GEO), a recognized public database that ensures data accessibility and longevity. The GSE number for our dataset is GSE262105 and it is freely accessible to the research community.

2. **Expansion of Table S2:** We have added a comprehensive list of individual genes based on the GESA analysis in the revised text, rather than just providing pathway summaries. This updated table offers a detailed overview of the gene expression profiles identified in our study, as presented in the revised manuscript's Table S2.

Thank you for your guidance in improving the accessibility of our research data, and we hope that these efforts will meet the journal's data sharing requirements.

4. Please indicate which phosphorylation site was measured for AKT in Figure 5, e.g., Thr308, Ser473, etc.

Reply Q4: Thank you for your insightful inquiries about the specific phosphorylation sites of AKT measured in our experiments. In our study, we specifically measured the phosphorylation at the AKT Ser473 site. The choice of Ser473 over Thr308 was informed by several considerations, grounded in both the context of cardiac research and findings reported in the literature.

Firstly, Ser473 phosphorylation has been widely recognized as a critical indicator for the full activation of the AKT pathway, particularly in cardiac tissues. Studies have shown that phosphorylation at Ser473 plays a pivotal role in cardioprotection, regulating processes such as cell survival, growth, and metabolism in response to stress and injury. For instance, the phosphorylation of AKT at Ser473 has been implicated in enhancing myocardial survival and reducing apoptosis under conditions of ischemic stress (*Circulation. 2001 Jul 17;104(3):330-5.*).

Secondly, the rationale for focusing on Ser473 rather than Thr308 in cardiac research stems from evidence suggesting that Ser473 phosphorylation precedes and may be necessary for Thr308 phosphorylation in the activation of AKT in cardiomyocytes. This sequential phosphorylation is thought to be crucial for the full activation of the AKT pathway, with Ser473 serving as a prerequisite for Thr308 phosphorylation (*Science. 2005 Feb 18;307(5712):1098-101.*). By targeting Ser473, our study aimed to capture the initial and essential step of AKT activation relevant to cardiac function and pathology.

Furthermore, previous studies have also supported the use of Ser473 phosphorylation as a reliable marker for the activation of the PI3K/AKT pathway in the heart. For example, research indicates that alterations in Ser473 phosphorylation are closely linked to changes in cardiac output and hypertrophic responses, reinforcing its relevance in cardiac physiology and disease (*Circulation. 2006 May 2;113(17):2097-104.*).

In summary, our selection of the Ser473 site was driven by its established significance in cardiac biology and the AKT activation process, supported by a body of research indicating its central role in cardiomyocyte survival and function. This choice reflects our aim to elucidate the mechanisms by which AKT phosphorylation contributes to cardiac remodeling.

Based on your suggestions, we have shown in manuscript that the phosphorylation of the AKT (Ser473) site being used. And thank you for helping us improve the quality of our manuscript.

Respond to Reviewer #2:

The study investigates the impact of Semaglutide (Sema) on cardiac remodeling induced by transverse aortic constriction (TAC) in mice. Sema is found to improve cardiac dysfunction, attenuate hypertrophy and fibrosis, and maintain mitochondrial morphology and function under chronic pressure overload conditions. Untargeted metabolomics analysis reveals positive effects on mitochondrial lesions, lipid accumulation, and ATP insufficiency. Transcriptional profiling identifies the

involvement of Creb5/NR4a1 in the PI3K/AKT pathway. Sema is shown to regulate myocardial energy metabolism and protect against pathological myocardial remodeling.

Major Issues:

1. The manuscript lacks clarity in explaining the rationale behind choosing the specific doses of Sema (4 µg/kg, 12 µg/kg, and 60 µg/kg) for the experiments. The justification for selecting the high dose (60 µg/kg) should be provided.

Reply Q1: Thank you for highlighting the need for clearer justification of the selected Sema doses (4 µg/kg, 12 µg/kg, and 60 µg/kg) in our study. The dosing strategy was developed from an exhaustive review of the existing literature and guided by outcomes from preliminary dose-response evaluations (*JACC Basic Transl Sci. 2018 Nov 21;3(6):844-857.*; *JACC Basic Transl Sci. 2023 Jul 26;8(10):1298-1314.*). Specifically, these doses were chosen to cover a broad spectrum of pharmacological responses, with the high dose (60 µg/kg) aimed at investigating the upper efficacy threshold and elucidating potential dose-dependent effects of Sema.

Notably, our findings revealed that different doses of Sema did not significantly impact blood glucose, cholesterol, or liver function in mice (**Figure S1D-G**), a result that aligns with the drug's primary use in clinical settings as a weight management solution (*Lancet Diabetes Endocrinol. 2024 Mar;12(3):184-195.*). This absence of significant side effects across the dosing range informed our decision to employ the high dose (60 µg/kg) for its notable impact on body weight, detailed in **Figure S1B-C**. This approach allows us to explore the drug's full therapeutic potential while ensuring safety and efficacy, particularly relevant for heart failure patients with obesity—a key patient demographic in clinical practice.

The insights gained from incorporating these doses contribute to a more nuanced understanding of Sema's pharmacodynamics. We are grateful for your constructive feedback, which has significantly enhanced the manuscript's clarity and overall quality.

Figure S1. Effects of different doses of Sema on body weight and glycolipids in mice that underwent TAC for eight weeks. (A) Mouse model intervention timeline diagram. (B-C) Effect of different doses of Sema on body weight of mice after sham or TAC surgery (n=6). (D) Effects on fasting blood glucose in mice treated with different doses of Sema every two weeks after sham or TAC surgery (n=6). (E) The results of the GTT after eight weeks of treatment with different doses of Sema (n=6). (F) Detection of the liver function indicators ALT and AST after eight weeks of treatment with different doses of Sema (n=12). (G) Measurement of total cholesterol and triglyceride levels after eight weeks of treatment with different doses of Sema (n=12). All results are shown as the mean±SEM, and analysis using one-way ANOVA followed by Bonferroni post hoc test (C and F-G) was conducted. For the analysis in (B and D-E), repeated measures ANOVA followed by Sidak post hoc test was conducted. *p* values are indicated. Source data are provided as a Source Data file.

2. The methods section does not elaborate on the criteria used for assessing blood glucose, cholesterol, and liver functions, raising concerns about the reliability of these measurements.

Reply Q2: Thank you for pointing out the necessity of elaborating on the criteria and methods used for assessing blood glucose, cholesterol, and liver functions. We understand the critical role that methodological clarity plays in ensuring the reliability and reproducibility of our findings. In response to your valuable comments, we have enhanced the methods section to include detailed descriptions of our assessment protocols as follows:

Methods section 2.5 Blood glucose measurements: In the modeled mice, the blood glucose levels in both standard and diabetic animals were determined using a kit reagent (Exactech blood glucose strip) according to the glucose oxidase method.

Methods section 2.6 Total cholesterol, triglycerides, alanine transaminase and aspartate transaminase measurements: Orbital venous blood was collected from

experimental mice and centrifuged for plasma. Enzymatic kit reagents were purchased from Nanjing Jiancheng Bioengineering Institute (Nanjing, China) and used to determine total cholesterol and triglycerides (TG). Serum concentrations of the liver enzymes alanine aminotransferase (ALT) and aspartate aminotransferase (AST) were determined by an automated biochemical analyzer (ADVIA® 2400 Siemens Ltd, Tarrytown, NY, USA) as previously described (*Cell Death Differ.* 2020 Feb;27(2):540-555.).

Thank you for your guidance in making our manuscript more methodologically rigorous.

3. While the study demonstrates the efficacy of Sema in reversing TAC-induced cardiac dysfunction and hypertrophy, the mechanistic insights into how Sema exerts these effects are somewhat limited. Further investigations into the specific signaling pathways and molecular mechanisms involved would enhance the manuscript's scientific depth.

Reply Q3: We sincerely appreciate your insightful feedback. Based on your suggestion, we have included the mechanism of the Sema reversal experiment as supplementary material and briefly mentioned it in the main text (**Figure S2**). The specific signaling pathways and molecular mechanisms regarding of Sema in reversal experiment were analysed as follows:

According to our previous studies myocardial remodeling has occurred after 4 weeks of TAC (*EBioMedicine.* 2022 Dec; 86:104359.; *Phytother Res.* 2023 May;37(5):1839-1849.). Sema ameliorates myocardial hypertrophy, fibrosis and cardiac dysfunction induced by TAC (**Figure 2**), and parallel transcriptomic analysis of myocardial tissue from mice in the reversal experiment revealed that the treatment of Sema significantly rescued TAC-induced changes at the gene level (**Figure S2A**). The reversal experiment showed a significant enrichment of DEGs in extracellular matrix (ECM), cell migration, and blood vessel development pathways (**Figure S2B**). We analyzed the sequencing data from the Sema reversal experiment and found that genes involved in the Creb5/NR4a1 pathway showed essentially trend as in the Sema treatment experiment (**Figure S2C, Figure 5B**). It has been demonstrated that ECM plays an important role in pressure overload-induced heart failure, and excessive deposition of ECM proteins leads to fibrosis and deterioration of cardiac function (*Circ Res.* 2022 Sep 16;131(7):620-636.).

Similarly, cell migration is crucial for fibrosis, with myofibroblasts under pressure overload conditions in heart failure showing enhanced migration abilities, secreting ECM proteins, and leading to myocardial remodeling and stiffening, thereby impairing cardiac function (*Biochim Biophys Acta Mol Cell Res.* 2024 Mar.). In addition, increased endothelial cell proliferation and promotion of angiogenesis significantly improve cardiac function in pressure overload-induced myocardial remodelling (*Arterioscler Thromb Vasc Biol.* 2024 Feb 8.). Sema exerts anti-fibrotic and anti-hypertrophic effects by attenuating ECM proteins (**Figure S2D**), and, consistently, Sema application inhibits deterioration of cardiac function, development of cardiac fibrosis and cardiac hypertrophy in response to pressure overload (**Figure 2**).

The regulation of cell migration is primarily mediated by the MAPK and ERK cascades (**Figure S2E**), with prior studies confirming their role in modulating myofibroblast migration and proliferation via NR4a1 expression (*Br J Pharmacol. 2023 Oct;180(19):2577-2598.*). Moreover, metformin's ability to ameliorate hyperglycemia-induced endothelial dysfunction by modulating NR4a1 (*Mol Pharmacol. 2021 Nov;100(5):428-455.*), suggests a close relationship between NR4a1 and vascular development. Additionally, NR4a1 is implicated in high-fat-associated endothelial dysfunction through the promotion of the CaMKII-Parkin-mitochondrial autophagy pathway in endothelial cells (*Cell Stress Chaperones. 2018 Jul;23(4):749-761.*), highlighting the importance of mitochondrial pathway regulation by NR4a1.

Our transcriptome analysis showed that the mechanism of Sema in the reversal experiment was mainly focused on the improvement of myocardial remodeling, which was closely related to the NR4a1 and Sema treatment groups. However, delving into the reversal mechanism of Sema was beyond the scope of our primary study focus. To keep our narrative coherent and focused, we confined the main body of our paper to discussing the mechanisms related to Sema treatment. Your constructive comments have significantly enriched our manuscript, deepening its scientific quality and depth.

Figure S2. Sema reverses cardiac hypertrophy, fibrosis and dysfunction in mice induced by TAC by regulating ECM deposition, cell migration, and vascular development pathways. (A). Heat map of gene level changes in the sema reversal experiment, n=3. (B). Gene Set Enrichment Analysis (GSEA) analysis network maps show key molecular pathways involved in the cardiac remodeling process and regulated by Sema. (C). Heat map showing key molecular signatures both involved in the cardiac

remodeling process and regulated by Sema, n=3. (D-F). Heatmap showing changes in genes involved in the ECM deposition, cell migration, and vascularization pathways involved and regulated by Sema during the reversal experiments, n=3.

4. The statistical analysis section needs improvement in reporting specific statistical tests used, including justification for choosing either homogeneity of variance or heteroscedastic data.

Reply Q4: Thank you for your constructive feedback. In response to your comments, we have revised the statistical analysis section to include detailed information on the statistical tests used, along with the rationale for their selection.

Justification for choice of tests: All data in this study are expressed as the mean \pm SEM, and statistical analysis was performed using GraphPad Prism 9.4.1 software. Guided by preliminary data analysis, comparisons between two groups with a normal distribution and homogeneity of variance were performed using an unpaired two-tailed Student's t test, whereas one-way analysis of variance (ANOVA) followed by Bonferroni post hoc test was conducted for comparisons among three or more groups. Analyses with $p < 0.05$ were considered statistically significant, and the corresponding p values are indicated above each legend. Related software and algorithms are listed in **Table S3**. We collected data from animal studies in a blinded manner and performed the final statistical analysis. In addition, we have written detailed statistical analyses in each of the figure legends sections.

We believe these revisions address your concerns and enhance the clarity. We appreciate the opportunity to improve our manuscript and look forward to your feedback.

5. The manuscript lacks a thorough discussion on potential limitations and challenges encountered during the study, such as any observed side effects of Sema or unexpected outcomes.

Reply Q5: We are grateful for your constructive suggestions. In response to your comments, we have revised the discussion section of our manuscript. Below are the main additions made to our manuscript.

In this comprehensive evaluation of Sema's impact on myocardial remodeling, our investigation employed a dose-response approach, administering 4.0, 12.0, or 60.0 mg/kg/day of Sema. The findings highlight a dose-dependent reduction in body weight across these doses without liver function compromise (**Figure S1**), with both the lowest and highest doses of Sema providing protective effects against pathological myocardial remodeling. Interestingly, the intermediate dose of 12.0 mg/kg/day was associated with diminished cardiac function, suggesting a nuanced relationship between Sema dosage and cardiac health that warrants further exploration to understand the underlying mechanisms fully. Moreover, the exclusive use of male mice limits the generalizability of these findings across genders, pointing to the need for gender-inclusive studies to fully ascertain the therapeutic potential of Sema.

Furthermore, while the Transverse Aortic Constriction (TAC) model is a robust method for simulating pressure overload-induced cardiac hypertrophy in rodents (*Circ Heart Fail.* 2009 Mar;2(2):138-44.), it falls short of capturing the multifaceted etiology of human cardiac disease, including genetic predispositions and susceptibility factors. This limitation emphasizes the importance of employing diverse experimental models to enhance our understanding of Sema's therapeutic implications in a broader, more clinically relevant context.

We are grateful for the opportunity to improve our manuscript based on your valuable feedback and hope that these changes address your concerns effectively. Thank you once again for your insightful comments.

6. Figures and tables could be better organized and labeled for improved readability. Additionally, some figures lack detailed captions that explain the significance of the data presented.

Reply Q6: Thank you for your suggestion. We have carefully reviewed the organization of our figures and tables to ensure that they follow a logical sequence consistent with the textual narrative. Each figure and table have been carefully relabeled with more descriptive captions. We have expanded the captions of the graphs to include detailed explanations of the data provided. These revised captions now provide a brief summary of what is depicted in each graph, and for complex figures we have included annotations. Thank you once again for your valuable feedback, which has been instrumental in improving the quality of our manuscript.

7. The introduction could benefit from a more comprehensive literature review, highlighting existing knowledge gaps that the current study aims to address.

Reply Q7: We greatly appreciate your insightful suggestion. We have made a detailed revision of our introduction as follows.

Preclinical studies have demonstrated that compounds promoting glucose oxidation, such as dichloroacetate (DCA), can inhibit pyruvate dehydrogenase complex (PDC) kinase. This inhibition stimulates PDC activity, enhancing glucose oxidation and mitochondrial respiration, and subsequently mitigates cardiac hypertrophy in HF models using animals (*Cardiovasc Res.* 2002 Mar;53(4):841-51.). Unfortunately, the toxic effects of DCA have limited its clinical use and further development (*Genet Metab.* 2021 Mar;132(3):211.). Another compound, Elamipretide, appears to influence metabolism by augmenting mitochondrial function; however, preclinical evidence indicates it fails to enhance cardiac function in HF scenarios (*J Card Fail.* 2020 May;26(5):429-437.). Consequently, identifying drugs that boost mitochondrial function and glucose oxidation to address the energy deficiency in HF represents a significant challenge (*Nat Rev Cardiol.* 2023 Dec;20(12):812-829.). Sema, in contrast, has exhibited cardiovascular advantages in clinical trials, notably among heart failure patients with obesity and preserved ejection fraction (*N Engl J Med.* 2023 Dec 21;389(25):2398.). Nevertheless, research into Sema's mechanistic effects on heart

failure remains scant. Our study seeks to bridge this knowledge gap by exploring the detailed metabolic mechanisms underlying Sema's cardiovascular benefits.

Thank you once again for your constructive comments.

Minor Issues:

1. The terminology used in the abstract, such as "glycolytic species products in the tricarboxylic acid cycle (TCA)," could be clarified for better understanding.

Reply Q1: Thank you for your suggestion, and we apologize for the misunderstanding due to an error in expression. We have revised the phrase "glycolytic species products in the tricarboxylic acid (TCA) cycle" to a more understandable term. The revised phrase now reads: "Untargeted metabolomics analysis revealed that Sema ameliorated mitochondrial lesions and reduced lipid accumulation and ATP insufficiency by promoting the **glycolytic product product, pyruvate, enters the tricarboxylic acid (TCA) cycle** after being converted to acetyl-CoA and increasing β -oxidation."

2. The introduction contains grammatical errors and could be refined for better readability.

Reply Q2: Thank you for your suggestion.

We have thoroughly reviewed the presentation to identify and correct all grammatical errors. Furthermore, we hired a professional agency to enhance the article. These changes have been highlighted in red for easy identification. We are grateful for your valuable feedback and the opportunity to improve our work.

AJE 
Editing Certificate

This document certifies that the manuscript

Semaglutide ameliorates cardiac remodeling by optimizing energy substrates utilization and through Creb5/NR4a1 axis.

prepared by the authors

Yu-Lan Ma^{1,2*}, Chun-Yan Kong^{1,2*}, Zhen Guo^{1,2*}, Ming-Yu Wang^{1,2}, Pan Wang^{1,2}, Fang-Yuan Liu^{1,2}, Dan Yang^{1,2}, Zheng Yang^{1,2}, Qi-Zhu Tang^{1,2#}

was edited for proper English language, grammar, punctuation, spelling, and overall style by one or more of the highly qualified native English speaking editors at AJE.

This certificate was issued on **August 27, 2023** and may be verified on the [AJE website](https://aje.com) using the verification code **50B1-1548-2F70-75BD-E07P**.

 Neither the research content nor the authors' intentions were altered in any way during the editing process. Documents receiving this certification should be English-ready for publication; however, the author has the ability to accept or reject our suggestions and changes. To verify the final AJE edited version, please visit our verification page at aje.com/certificate. If you have any questions or concerns about this edited document, please contact AJE at support@aje.com.

AJE provides a range of editing, translation, and manuscript services for researchers and publishers around the world. For more information about our company, services, and partner discounts, please visit aje.com.

3. The rationale for choosing the 8-week time point for the experiments needs clarification. Was this duration based on previous studies or empirical observations?

Reply Q3: Thank you for your insightful query. The choice of the 8-week time point for our experiments was carefully considered and is supported by both empirical observations and a review of relevant literature. This duration aligns with the time frame commonly observed for the development of significant cardiac remodeling and dysfunction following Transverse aortic constriction (TAC) surgery, as reported in previous studies.

Transverse aortic constriction (TAC) is a widely recognized animal model for studying pressure overload-induced cardiac hypertrophy. This model illustrates a progression from compensatory left ventricular growth to pathological states including capillary rarefaction, cardiomyocyte death, and fibrosis, culminating in heart failure (*Circ Heart Fail.* 2009 Mar;2(2):138-44.). Early post-TAC phases feature significant inflammation, with a notable increase in inflammatory cells within two weeks (*Circulation.* 2019 Aug 6;140(6):487-499.). By four weeks, observable myocardial remodeling and functional impairment manifest (*EBioMedicine.* 2022 Dec; 86:104359.), advancing to end-stage heart failure by eight weeks (*Metabolism.* 2024 Feb 17; 154:155818.; *Research (Wash D C).* 2024 Feb 6; 7:0303.; *Front Physiol.* 2022 Mar 7;13:777284.).

Our study aimed to examine Sema's effects across the full spectrum of pathological changes induced by pressure overload, encompassing both the remodeling phase and the transition to heart failure. Echocardiographic assessments at four weeks post-TAC indicated that Sema improved cardiac function during the remodeling phase (**Figure R1**). However, our primary focus was to understand Sema's drug effects and mechanisms at the late-stage pathological changes, specifically at 8 weeks post-TAC.

We appreciate the opportunity to clarify this aspect of our study design and thank you for your valuable feedback, which has undoubtedly contributed to the rigor and clarity of our research.

Figure R1. Sema attenuates cardiac dysfunction in mice that underwent TAC for four weeks. (A) The results of the EF after four weeks of treatment with Sema (n=6). (B) The results of the FS after four weeks of treatment with Sema (n=6). All results are shown as the mean±SEM, and analyzed using an unpaired two-tailed Student's t test. *p* values are indicated.

4. The explanation of the methods for echocardiographic analyses, specifically the M-mode tracer method, requires more detail to ensure reproducibility.

Reply Q4: Thank you for your valuable comments. A more detailed description of the echocardiographic methods, particularly the M-mode, is provided in revised manuscript as below.

Echocardiography was performed at 4 or 8 weeks after TAC. Refer to our previous study (*Circulation. 2024 Feb 27;149(9):684-706.; EBioMedicine. 2022 Dec; 86:104359.*), echocardiography was analyzed with a 10-MHz linear array ultrasound transducer equipped with a 30-MHz probe (Vevo 3100 system Visual Sonics). First, the mice were anaesthetised with a mixture of 4% isoflurane and 0.5 L/min oxygen. Conductive adhesive was applied to the copper plate of the physiological information monitoring table and the paws of the mice were secured to obtain electrocardiogram (ECG) and respiratory information. The probe was positioned in the parasternal short-axis view and M-mode ultrasound images were acquired for diastolic and systolic left ventricular internal diameters (LVIDd and LVIDs). Multiple cardiac cycles were tracked and traced along the endocardial (Endo) trajectory of the anterior wall of the LV, and then the Endo of the posterior wall of the LV was similarly traced. On this basis, the following calculated values can be obtained: LV end-diastolic volume (EDV), LV end-systolic volume (ESV), stroke volume, ejection fractions (EF) and fractional shortening (FS). Notably, the LVFS was calculated by the formula: $LVFS = (LVIDd - LVIDs) \times 100 / LVIDd$. The LVEF was also evaluated using the Teichholtz formula: $LVEF = ([100 - Y] \times 0.15) + Y$; $Y = (LVIDd^2 - LVIDs^2) \times 100 / LVIDd^2$. To better answer your questions, the specific operations of mouse ultrasound and the detailed analysis of the data are shown in the chart below (**Figure R2**).

Thank you once again for your constructive comments, which have been instrumental in improving the quality and reproducibility of our research.

Figure R2. Schematic diagram of mouse ultrasound manipulation and ultrasound data analysis. (A) The schematic diagram of ultrasound manipulation in mice. (B) The representative schematic diagram of the analysis of ultrasound data.

5. The authors should provide more information on the potential limitations of the study, especially regarding the TAC model and its relevance to human cardiac pathology.

Reply Q5: We greatly appreciate your insightful feedback. We recognize that although the TAC model is a well-established party method for inducing pressure overload and subsequent cardiac hypertrophy in rodents (*Circ Heart Fail.* 2009 Mar;2(2):138-44.; *Circ Res.* 2024 Feb 16;134(4):393-410.; *Signal Transduct Target Ther.* 2024 Feb 19;9(1):45.), it does not fully replicate the complex etiology of human heart disease. The model focuses primarily on mechanical pressure overload, may not fully encompass the multifactorial nature of human cardiac pathology, including genetic predisposition, environmental factors, and systemic diseases such as diabetes and hypertension. The TAC model has greatly facilitated cardiac physiological assessment with new technologies such as ultra-high resolution ultrasound analysis (**Vevo 3100 system Visual Sonics**). Although it does not fully mimic human cardiac pathology, it is important for some "proof-of-principle" purposes, such as identifying important gene or protein targets or beneficial drugs. This could pave the way for the development of new molecular or drug therapies (*Eur Heart J.* 2022 Dec 1;43(45):4739-4750.; *Circ Res.* 2024 Feb 16;134(4):393-410.).

In response to your suggestion, we have revised the limitation sections of the study in the manuscript. Thank you once again for your constructive feedback.

6. There is a need for clarification on how the study controlled for potential confounding factors in the TAC model, such as variations in surgical techniques or individual mouse responses.

Reply Q6: Thank you for emphasizing the need to clarify how our study controls for potential confounders associated with the transverse axis aortic constriction (TAC) model. Below are the key steps taken.

To minimize variations in surgical technique, all TAC procedures are performed by specialized surgical staff who have helped the team publish several representative articles based on the TAC model (*EBioMedicine.* 2022 Dec;86:104359.; *Cell Rep Med.* 2023 Dec 19;4(12):101334.; *Nat Commun.* 2023 Aug 16;14(1):4967.). This approach ensured consistency in the application of contraction and minimized the variability in the degree of pressure overload. To ensure uniform ligation of the aorta in all subjects, all instruments were consistent throughout the procedure, including the needles and thread gauges from the same batch. Wherever possible, male mice of uniform age and body weight from the same litter source were used for the surgical model, and the mice were grouped in a randomized and blinded manner. Mice were maintained by the same caregivers throughout the experiment, and the housing environment was maintained consistently (including temperature, humidity, light and catering). Sample sizes were calculated based on preliminary data to ensure sufficient power to detect significant differences in the case of individual differences. By implementing these measures, we

aim to ensure that our findings are robust and reflect the effects of TAC, rather than surgical variability or artifacts of individual mouse responses.

Thank you again for your constructive feedback, which greatly enhances the rigor and transparency of our research methods.

7. In the discussion, the authors could provide a more comprehensive comparison of their findings with existing literature to underscore the novelty and significance of their results.

Reply Q7: We appreciate your valuable feedback. In response to your suggestion, we have extensively revised the discussion section. The changes were highlight in red. We are grateful for the opportunity to enhance our manuscript with this comprehensive analysis, and we hope that these changes address your concerns effectively.

Respond to Reviewer #3:

In this study the authors have investigated the effects of the GLP-1R agonist Semaglutide (Sema) in cardiomyocytes and a transverse aortic construction surgery as the disease model. The authors have presented an extensive data set showing that the protective effects of Sema involve the Creb5/NR4a1 axis.

1. The authors convincingly show in the TAC model that Sema attenuates markers of hypertrophy, fibrosis and dysfunction (mitochondrial). Did the authors determine levels of ROS and effects Δ MMP in these studies?

Reply Q1: Thank you for your inquiry regarding about the effects of Sema on reactive oxygen species (ROS) levels and on mitochondrial membrane potential (Δ MMP).

Sema as a weight loss drug regulates energy metabolism and mitochondria acts as the centre of energy metabolism. Therefore, oxidative stress-related indicators were detected at the early stage of our experiment. The results showed that Sema attenuated the TAC-induced increase in MDA and GSH levels (**Figure R3A**). Moreover, superoxide dismutase 2 (SOD2) western blotting showed that Sema rescued the reduction of SOD2 induced by TAC (**Figure R3B**). However, multi-omics data showed that Sema is involved in mitochondria-associated glycolipid metabolism, and pathological trails suggested a relationship between Sema and mitochondrial morphological integrity (**Figure 3~5**).

Based on this, we focused on mitochondrial morphology and substrate metabolism rather than mitochondrial free radical production and scavenging, which can be seen as the upstream of oxidative stress. Therefore, we did not include oxidative stress-related data in the final manuscript. We agree that Δ MMP is an important indicator of ROS levels and regret that we were unable to provide relevant data. Our results suggest that Sema may play a regulatory role in the ventricular myocardium by affecting acetylcholine levels through the influence of P cells (*Nature. 2023 Jul;619(7971):801-810.*) (**Figure S8**), and that Sema might not directly affect ventricular myocytes and could not mimic the therapeutic effects in vitro. Furthermore, the complexity and inevitable false positives of detecting Δ MMP at the tissue level contribute to the ultimate failure of Δ MMP detection. This may be a limitation of the study in this article. Although Δ MMP was not detected, we experimentally verified the mitochondrial morphology, fusion fission-related proteins, respiratory chain-related gene levels and oxygen consumption rate in our study (**Figure 3, Figure 7**) and provided alternative data below indicating ROS levels. We hope these have answered your concerns about ROS level detection. Thank you again for your constructive feedback.

Figure R3. Sema attenuates the level of oxidative stress induced after 8 weeks of TAC. (A) Quantitative results of myocardial MDA and GSH levels ($n=6$). (B) Western blot analysis and quantification of SOD2 in the cardiac tissue of Sema-treated mice 8 weeks after sham or TAC surgery and quantification levels were normalized to β -actin ($n=6$). All results are shown as the mean \pm SEM. and analysis using one-way ANOVA followed by Bonferroni post hoc test was conducted. p values are indicated.

2. The interactions of Sema, TAC, Creb5 in the disease model are of interest and need some explanation: The reference #39 which identify Creb5/NR4A1 as an important downstream pathway of P13K/AKT mentions the AKT-Creb5 connection but not NR4A1 and reference #40 is primarily focused Nur77-dependent regulation of microRNAs. Are there more relevant references that should be included? The results suggest that TAC induces nuclear export of NR4A1, and this is inhibited by Sema and may be important for TAC-induced mitochondrial damage. This should be confirmed using a nuclear export inhibitor such as leptomycin B. In addition, the changes in NR4A1 phosphorylation can be important for inducing NR4A1 nuclear export. The authors also state “Transcriptional profiling revealed that Sema exerted its effects on myocardial energy metabolism by regulating Creb5/NR4a1 in the canonical PI3K/AKT pathway. Specifically, Sema reduced the expression of NR4a1 and its translocation from the nucleus to mitochondria.” This suggests that TAC induces both extranuclear (export) and nuclear pathways (inhibited by Sema) and therefore the balance of NR4A1 (nuclear/NR4A1 (mitochondrial)) may be required for an optimal “damage” response and this needs to be addressed.

Reply Q2: Thank you for your insights into the relationship between the roles of Creb5 and NR4a1 and the regulation of NR4a1 by Sema. In response to your query, we have carefully considered your suggestion. The specific answers are as follows.

We have revised the manuscript and added appropriate references to clarify the relationship Creb5, NR4a1 and the P13K/AKT pathway in section 3.5. Creb5 is a crucial component of the downstream signaling pathway of PI3K/AKT (*Front Pharmacol. 2022 Jun 9;13:862709.*). It is a member of the CREB (cAMP-responsive element-binding protein) family of proteins (*Oncol Lett. 2017 Dec;14(6):8156-8161.*). Studies have demonstrated that the CREB family plays a regulatory role in modulating the activity of NR4a1 (*J Neuroinflammation. 2019 Oct 28;16(1):192.*). We further validated the analysis of PI3K/AKT pathway-related DEGs using RNA-seq data and found that Sema significantly reduced TAC-induced activation of the transcription factors Creb5 and NR4a1 (**Figure 5B**). We concluded that Creb5 may act as a downstream of P13K/AKT to regulate NR4a1, which was also confirmed in our study (**Figure 5E**).

As for NR4a1 nuclear translocation inhibition experiments: We believe that the use of nuclear export inhibitors (e.g. leucovorin B) to validate NR4a1 nuclear translocation is highly plausible. However, since the present manuscript shows that Sema may affect the release of acetylcholine through its action on P cells to exert a regulatory effect on the left ventricle (*Nature. 2023 Jul;619(7971):801-810.*) (**Figure**

S8). It is not possible to mimic the therapeutic effects of Sema by the cellular assay in vitro.

As for the expression and translocation of NR4a1: We extracted total tissue proteins, nuclear proteins and mitochondrial proteins to validate the regulation of NR4a1 by Sema. The results showed that NR4a1 phosphorylation levels (p-NR4a1) were increased by TAC and were indeed diminished by Sema treatment (**Figure R4A:** see in Figure 5C in revised manuscript). Previous studies found that YAP knockdown downregulates the phosphorylation level of NR4a1 in hela cells, and induces nuclear export and mitochondrial co-localization with NR4a1 (*Cell Rep. 2020 Oct 20;33(3):108284.*). However, cisplatin induced increased levels of NR4a1 phosphorylation while promoting NR4a1 nuclear export in human umbilical cord mesenchymal stem cells (*Reprod Biol Endocrinol. 2022 Aug 19;20(1):125.*). Our results found the phosphorylation levels of NR4a1 elevated by TAC. Meanwhile, increased level of NR4a1 was found to be localized outside the nucleus. Thus, we suggest that elevated levels of NR4a1 phosphorylation might induce NR4a1 nuclear export in cardiomyocytes. Moreover, we examined NR4a1 (nuclear) and NR4a1 (mitochondrial) protein levels and counted the ratio of NR4a1 (nuclear/mitochondrial). The results revealed that TAC induced a synchronized elevation of NR4a1 in the nucleus and mitochondria (**Figure R4B:** see in Figure 5D in revised manuscript). However, the ratio of nuclear to mitochondrial NR4a1 was decreased (**Figure R4C:** see in Figure 5D in revised manuscript). Sema application increased the ratio of nuclear to mitochondrial NR4a1, suggesting that Sema reduced NR4a1 expression while decreasing its translocation to mitochondria. We have clarified and supplemented this section in the revised manuscript (see in Figure 5C, D in revised manuscript).

We hope that our explanations and experimental validations above have answered your questions. Thank you again for your constructive feedback.

Figure R4. Regulation of NR4a1 expression and translocation and phosphorylation by Sema. (A) Western blot analysis and quantification of p-NR4a1 in the cardiac tissue of Sema-treated mice 8 weeks after sham or TAC surgery and quantification levels were normalized to β-actin (n=6). (B-C) Western blot analysis and quantification of NR4a1 in mitochondrial and nuclear proteins and quantification levels were normalized NR4a1 (nuclear/mitochondrial) (n=6). All results are shown as the mean±SEM, and analysis using one-way ANOVA followed by Bonferroni post hoc test was conducted. *p* values are indicated. Source data are provided as a Source Data file.

3. Do the authors have any idea how Sema, working through the GLP-1R assay affects NR4A1 expression and function and intracellular location? It should also be

noted that NR4A1 ligands can also modulate receptor-dependent glucose metabolism and fibrosis and this should be integrated into the Discussion.

Reply Q3: Thank you for your insightful inquiry. In response to your comments, we have revised our manuscript to include the effects of NR4a1 ligands on glucose metabolism and fibrosis. Below, we will outline how Sema, working through the GLP-1R assay affects NR4a1 expression and function and intracellular location.

Single-cell drug-targeting maps predict that Sema has the strongest effect on P cells compared with other cardiac cells. It was found that P cells promote autonomic activity through nerve growth factor (NGF) signaling (*Nature*. 2023 Jul;619(7971):801-810.). Acetylcholine released from cardiac vagal autonomic nerves acts primarily on myocardial M muscarinic receptors. In the cardioprotection afforded by remote ischemic conditioning, the vagus nerve stimulates secretion of GLP-1 from intestinal peptides, which then activates GLP-1R to reduce the size of myocardial infarcts through a mechanism involving the M muscarinic receptor (*Basic Res Cardiol*. 2021 May 3;116(1):32.). M muscarinic receptor activation has been shown to improve cardiac function in a variety of cardiac diseases (*Life Sci*. 2023 Sep 15;329:121971.; *Proc Natl Acad Sci U S A*. 2023 Jul 11;120(28):e2210152120.; *Basic Res Cardiol*. 2023 Oct 6;118(1):43.). M muscarinic receptors have been shown to modulate the PI3K/AKT pathway (*Proc Natl Acad Sci U S A*. 2023 Apr 4;120(14):e2212476120.; *J Mol Histol*. 2024 Feb;55(1):51-67.; *J Cell Physiol*. 2015 Apr;230(4):767-74.). Therefore, we hypothesized that the protective effect of Sema on myocardial remodeling might also be related to the activation of M muscarinic receptors. We performed M muscarinic receptor blockade experiments with Atropine and found that the ameliorative effect of Sema on cardiac function was significantly blocked by Atropine (**Figure S8**). Taken together, we suggest that Sema may exert its cardioprotective effect by activating GLP-1R expressed by P cells, triggering the release of acetylcholine, and modulating ventricular function through activation of M muscarinic receptors. Although this conclusion is not supported by exhaustive experiments, these findings still provide the possibility that Sema can modulate ventricular muscle through acetylcholine, with a possible pathway for M muscarinic receptors to influence Creb5/NR4a1 expression through modulation of PI3K/AKT.

Structural studies of the ligand-binding domain of NR4a1 have shown that there are no suitable ligand-binding cavities for binding complex ligands, and NR4a1 has no known endogenous ligand (*J Diabetes Res*. 2018 Jun 14;2018:9363461.; *Cells*. 2019 Nov 1;8(11):1373.). However, several recent studies have identified structurally diverse compounds that bind and activate or inactivate nuclear NR4a1 (*J Exp Clin Cancer Res*. 2021 Dec 14;40(1):392.). Cytosporone B (Csn-B) is a naturally occurring agonist for Nur77 (NR4a1) used to cancer and fibrosis-related diseases (*Nat Chem Biol*. 2008 Sep;4(9):548-56.; *Mol Med*. 2023 May 9;29(1):63.). Studies have shown that NR4a1 function is antagonized by the chemical compound ethyl 2-[2,3,4-trimethoxy-6-(1-octanoyl) phenyl] acetate (TMPA) in diabetic mice, implicates Nur77 (NR4a1) as a new and amenable target for the design and development of therapeutics to treat metabolic diseases (*Nat Chem Biol*. 2012 Nov;8(11):897-904.). Although these

structurally diverse compounds that activate or inactivate nuclear NR4a1, there is no study that proves they can modulate the nuclear transcription and translocation levels of NR4a1 in cardiovascular disease. It is not clear whether these compounds are over-activated or over-inhibited. Moreover, it is important to keep NR4a1 in a dynamic equilibrium. This is also the innovation of the mechanism of NR4a1 regulation by Sema in this study, and Sema also has the effect of weight loss, which is expected to be an effective clinical drug for patients with obesity cardiomyopathy by regulating metabolism with NR4a1 as the target. By addressing these issues, we aim to provide a more nuanced understanding of the underlying mechanisms by which Sema exerts its cardioprotective effects and the role of NR4a1 in mediating these effects.

Thank you again for your constructive comments, which have greatly improved the depth and quality of our manuscript.

Figure S8. Cardioprotection induced by Sema is mediated by a muscarinic mechanism. (A) Effect of Atro on body weight of mice after TAC surgery, n=12. (B-E) Cardiac function of mice after TAC 4W (EF, FS, EDV and Stroke Volume). (F) The Cardiac output of mice (n=12). (G-H) Cardiac function of mice after TAC 8W (EF and FS). All results are shown as the mean±SEM, and analysis using one-way ANOVA followed by Bonferroni post hoc test (A-F) was conducted. For the analysis in (G and H), using an unpaired two-tailed Student's t test. *p* values are indicated. Source data are provided as a Source Data file.

REVIEWERS' COMMENTS

Reviewer #1 (Remarks to the Author):

The authors have been very responsive to both my and the comments of the other reviewers. The manuscript has been further improved and I have no additional comments.

Reviewer #2 (Remarks to the Author):

The authors have adequately addressed most of the concerns raised, significantly improving the manuscript's clarity and quality.

Reviewer #3 (Remarks to the Author):

The authors have made extensive revisions based on critiques by all of the Reviewers. While they did not answer every question entirely they provided some rational explanations. This paper will contribute to the field of NR4A1 biology and I recommend acceptance.